# STAND: Self-Aware Precondition Induction for Interactive Task Learning

Daniel Weitekamp [1]   Glen Smith [1]   Kenneth Koedinger [2]   Christopher MacLellan [1]

## Abstract

In interactive task learning (ITL), AI agents learn new capabilities from limited human instruction provided during task execution. STAND is a new method of data-efficient rule precondition induction specifically designed for these human-in-the-loop training scenarios. A key feature of STAND is its self-awareness of its own learning—it can provide accurate metrics of training progress back to users. STAND beats popular methods like XG-Boost, decision trees, random forests, and version spaces at small-data precondition induction tasks, and is highly accurate at estimating when its performance improves on holdout examples. In our evaluations, we find that STAND shows more monotonic improvement than other models with low rates of error reoccurrence. These features of STAND support a consistent training experience, enabling human instructors to estimate when they have finished training and providing active-learning support by identifying trouble spots that require more training. STAND achieves this by efficiently learning a compact space of greedy classifiers consistent with training data, rather than a finite ensemble of alternatives.

## 1. Introduction

This work introduces STAND and applies it to precondition induction tasks within Interactive Task Learning (ITL) systems. ITL broadly envisions AI systems capable of learning open-ended capabilities entirely from human instruction (Laird et al., 2017). While most AI systems improve task performance by learning or planning in pre-built environments or by training on static datasets, ITL systems learn novel concepts, actions, and broad capabilities directly from human instruction. ITL is a broad area of human-in-the-loop machine learning, oriented toward situations

where AI agents need to be taught on the fly, or where non-programmers want to teach agents robust behaviors through natural instruction that cannot be achieved robustly with data-driven machine learning or pretrained generative AI.

ITL systems have supported various instructional modalities, including learning from observation and demonstration (i.e., programming-by-demonstration), direct manipulation of knowledge structures via user-interfaces (Li et al., 2020), learning from natural language instruction (Lawley & Maclellan, 2024), or using multiple modalities in combination (Woodward et al., 2020; Allen et al., 2007; Kirk & Laird, 2019). In systems that learn production rules or hierarchical rules (which we focus on in this work), a distinction can be drawn between the elements of learned rules that carry out tasks (i.e., effects) and the elements that gate their execution (i.e., preconditions). The former is often simpler to demonstrate or articulate. For instance, one could teach an agent a rule that carries out a series of actions in a game by demonstrating them or saying them. Yet, articulating the preconditions for those actions is more challenging, both because it can be hard to anticipate how individual rules are selectively applied across edge cases of the intended behavior, and because the environment's internal state representation may be invisible or too complex to feasibly display to users. A more user-friendly approach, which we evaluate STAND on here, is to learn rule preconditions via interactive inductive training: users grade the actions that result from applying learned rules as correct or incorrect, and these labels are used to induce preconditions.

We evaluate STAND[1] on precondition induction scenarios within two different ITL systems that learn rules in the form of Hierarchical Task Networks (HTNs) (Nau et al., 1999): (1) VAL is an ITL system where users can build HTN methods by instructing agents about how to decompose tasks into subtasks, and has been applied in the context of game-based tasks (Lawley & Maclellan, 2024). (2) AI2T (Weitekamp et al., 2024) is a system for authoring-by-tutoring (Weitekamp et al., 2020 in press; MacLellan & Koedinger, 2020; Matsuda et al., 2015) where tutoring systems are built by interactively demonstrating problem solutions to an agent and grading its step-by-step solutions on subsequent problems. In AI2T, the breakdown of high-level tasks into HTN meth-

[1]Georgia Institute of Technology [2]Carnegie Mellon University. Correspondence to: Daniel Weitekamp <weitekamp@gatech.edu>.

*Proceedings of the 43rd International Conference on Machine Learning*, Seoul, South Korea. PMLR 306, 2026. Copyright 2026 by the author(s).

---

[1]https://github.com/DannyWeitekamp/STAND

ods (subtask sequences) is learned by bottom-up induction from demonstrated action sequences. In both of these systems, HTN subtask decompositions are readily learned from just a few examples, yet successful precondition induction often requires dozens of positive and negative examples.

Precondition induction in these scenarios requires operating effectively under the following task characteristics:

1. Training data is **small** and **heavily imbalanced** since negative examples attenuate as performance improves.

2. Target preconditions are **logical statements**, not weighted or nonlinear combinations, thus the useful features are **tabular**, and often **categorical** in nature.

3. The feature space is often **large**, and full of distractor features that produce spurious correlations.

The human-in-the-loop nature of this training paradigm imposes additional desirable features for the precondition induction system:

1. Instructors can benefit from an **estimate of the agent's progress toward learning correct logical statements**.

2. Likewise, the AI should be an **active learner**, and estimate which kinds of examples will aid its learning.

3. Also, **monotonic learning** is helpful: performance, and certainty estimates should strictly increase, with new examples strictly fixing errors and not causing new ones.

Prior work on authoring-by-tutoring has highlighted cases where error reoccurrence can cause issues (Weitekamp et al., 2020 in press; 2021). Consider a system that has learned several rules with still incomplete preconditions. One set of preconditions may correctly halt the execution of a rule in state A, but when the user grades a new instance of applying the rule in state B, the refit preconditions may incorrectly permit applying the rule in state A. The user could return to state A (or a similar one), only to find a new erroneous action that had been previously ruled out (a false positive reoccurrence). In the worst case, they may be unable or unaware that they should return to state A—missing an opportunity to provide negative feedback and improve performance.

We illustrate that STAND succeeds in each of the roles listed above. STAND learns quickly from little data and has low rates of error reoccurrence as new examples are collected. STAND's predicted label probabilities on unseen examples tend to increase monotonically toward their true label, and these probabilities are accurate estimates of hold-out set precision, especially for high prediction probabilities ($> 95\%$). In an ITL setting, human instructors can interpret STAND's high prediction probabilities as an indication of training completion, and low probabilities as an active learning guide for identifying important edge cases.

## 2. Related Work

Many well-established methods inductively learn concepts and rules from examples. STAND works on top of greedy divide-and-conquer (Breiman et al., 2017) and sequential covering (Quinlan, 1996) approaches, and borrows ideas from version space learning (Mitchell, 1982), to build an approximate version space structure over disjunctive normal concepts—a representation language that is intractable to learn under the typical candidate elimination algorithm (Hirsh, 1992). Workarounds like IVSM (Hirsh, 1994), DiVS (Sebag, 1996), and VSSM (Hong & Tseng, 1999) have been proposed to sidestep version space collapse when applied to disjunctive and noisy domains. Other methods have extended the notion of version spaces to recursive grammars (VanLehn, 1987), support vector machines (Tong & Koller, 2001), and programming-by-demonstration-based for text editing (Lau et al., 2003).

Inductive logic programming (ILP) methods focus on learning short relational logical programs within compelling, yet highly curated domains, often with few distractor features. ILP approaches often search over candidate logical programs, using predicate invention, meta-rule search, prior knowledge constraints, and other techniques (Cropper & Dumančić, 2022). Like other decision tree variants, STAND is a propositional learner. However, when applied within VAL and AI2T, relative featurization (Weitekamp et al., 2025a) is used to re-express grounded feature predicates relationally in terms of variables from partially learned rules. In principle, the STAND approach is also amenable to more explicit ILP extensions, such as TILDE, a tree-based ILP method (Blockeel & De Raedt, 1998).

Like few-shot learning (Song et al., 2023) and meta-learning (Gharoun et al., 2024), ITL learns from limited data. But, while these and related approaches tune pre-trained foundation models (Hu et al., 2022), or guide behavior with in-context examples (Agrawal et al., 2023), ITL systems learn novel, interpretable knowledge structures bottom-up from user instruction (Kirk & Laird, 2019), often with an expectation that users can train agents to exhibit behavior that is 100% accurate within a well-defined scope. As in interactive machine learning (Fails & Olsen Jr, 2003) where users respond to model predictions, ITL systems often, as in this work, signal knowledge formation or learning progress back to the user (Guo et al., 2022). STAND learns preconditions to complete partially induced HTN methods intended for reactive control. This differs from HTN precondition learning in planning domains that reduce search time over verifiable plans (Langley, 2025).

In this ITL training context, a key feature of STAND is its ability to provide accurate estimates of its performance on unseen examples. While statistical averaging among ensemble predictors like random forests often shows good calibra-

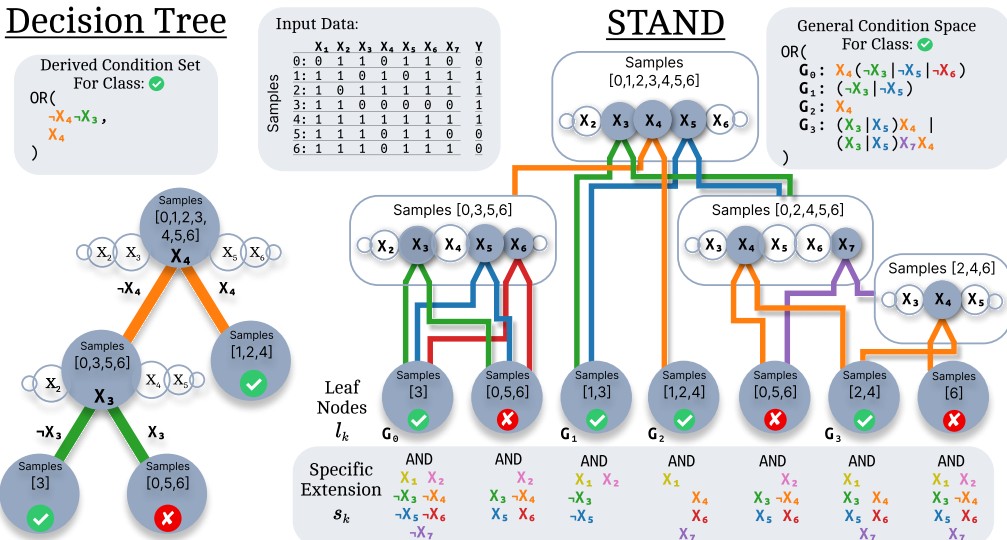

Figure 1. Decision tree and STAND fit to the same input data. In STAND, multiple splits (filled grey circles) are expanded per node. STAND builds a general condition space (top-right) that is bounded below by a specific extension (bottom).

tion to holdout accuracy in large-data settings (Niculescu-Mizil & Caruana, 2005), we show that the missing information in small-data induction requires a more explicit engagement with the ambiguity of learning good generalizations from limited data. The Know What it Knows (KWIK) framework (Li et al., 2008) is a kindred example of an explicit treatment of ambiguity, rather than simple statistical averaging, applied to transition function estimation in reinforcement learning. By contrast, STAND captures ambiguity by maintaining a space of all consistent classifiers. This differs from approaches like TreeFARMS (Xin et al., 2022) and SPLIT (Babbar et al., 2025), which enumerate good tree classifiers within the Rashomon set of almost-optimal models for inspection or optimization. Instead of enumerating generalizations individually, STAND produces predictions and useful metrics directly from its compact representation of all good solutions. This makes learning and inference with STAND efficient for low-latency interactive training.

## 3. STAND

In ecology, a stand is a contiguous region of trees that share similar characteristics. Analogous to its namesake concept, the STAND algorithm learns a compact collection of classifiers embedded in a shared data structure that holds every classifier that would be generated by a randomized greedy learning process, plus a space of extensions on those classifiers. Here, we describe STAND over decision trees, although the same method can be applied to sequential covering (described in Appendix A). Similar to a version space (Mitchell, 1978; 1982), STAND builds a space of generalizations consistent with training data enclosed by two boundary sets: a general set $G$ and a specific set $S$.

### 3.1. Constructing the General Set

STAND's general set G is constructed by exploring more possibilities than a typical greedy approach, like decision trees. Typically, greedy algorithms incrementally build on a working set of selection criteria $t_{L-1}, ..., t_0$ (e.g., a path from root to a node in a decision tree), by selecting a single best next literal $t_L$ that maximizes some gain function $F(X, Y)$ (e.g., decreases in entropy or Gini impurity).

$$t_L = \arg \max_{t(X_j)} F(X_j, Y_k | t_{L-1}, ..., t_0) \quad (1)$$

STAND instead picks the set of literals (i.e., predicates or their negations) $T_L$ whose gain value is within some extra acceptance range $\alpha$ ($\alpha = 0.1$ works well) of the maximum $M$. The value of $\alpha$ can be fixed or attenuated dynamically as a function of the amount of training (see Appendix A).

$$M = \max_j F(X_j, Y_j | t_{L-1}, ...) \quad (2)$$
$$T_L = \{t(X_j) : F(X_j, Y_j | t_{L-1}, ...) \geq M(1 - \alpha)\} \quad (3)$$

Variants of decision trees that hold multiple *nearly best* feature splits instead of just one are called option trees (Kohavi & Kunz, 1997). The number of ties in an option tree tends to be greatest in small data settings and in deep nodes that select few training examples. Without taking extra measures, this structure can easily blow up exponentially, overwhelming available memory. Lazy expansion methods like Hoeffding trees (Pfahringer et al., 2007) avoid this explosion by waiting to expand until more data is collected. By contrast, STAND expands every option exhaustively and avoids combinatorial explosion by caching each node by the subset of samples that it selects. Since the set of expanded splits at each node (and its resulting subtree) depends en-

tirely on the training samples selected by that node, any outgoing edges that select the same subset of samples can be routed to the same shared node. This forms a lattice that acts as STAND's general set G, allowing multiple paths from the root to the same leaf (see Figure 1).

Figure 1 shows an example of a decision tree, and STAND fit to the same data. Each sample 0-6 in the training data has seven binary features $X_1, ..., X_7$. In the decision tree, $X_4$ is selected randomly at the root node from among the best features for splitting the data. $X_4$ (i.e., $X_4 = 1$) selects a pure subset of three positive samples, and $\neg X_4$ (i.e., $X_4 = 0$) selects an impure subset that is then split further by $X_3$. By contrast, STAND splits $X_4$ at the root, and also $X_3$ and $X_5$. STAND's node caching trick makes the 6 edges formed by these 3 splits lead to only 4 nodes (two of which are reused by two edges) instead of 6 nodes (one per edge) like in an option tree. The 4 nodes downstream of the root are reached by following edges that select sample sets [0,3,5,6], [1,3], [1,2,3], and [0,2,4,5,6] respectively. The middle two are leaves because they only select positive samples, while the others are still impure and are further split into pure leaves. Note that in STAND (unlike decision trees), a single sample can filter into multiple leaves. For instance, both the first [3] and third leaf [1,3] include sample 3. By reusing nodes, STAND learns a complete space of best candidate decision trees, often in only slightly more time than it takes to learn a single decision tree.

## 3.2. Constructing the Specific Set

STAND's specific set $S$ is made up of specific extensions $s_k$ for each leaf $k$. For categorical features, each $s_k$ consists of the set of features that have the same value for all training examples in leaf $k$. For all continuous-valued features, $s_k$ holds the range of their minimum and maximum values. When making predictions with STAND, $S$ provides additional criteria for evaluating new examples beyond the top-down filtering criteria of a typical greedy classifier.

## 3.3. The Specific Contribution to Label Certainty

Since $G$ is a lattice, an individual example $\mathbf{x}$ may filter into multiple leaves, and those leaves can differ by label prediction $y = c$. $\text{Cert}(y = c|\mathbf{x}) \in [0, 1]$ is the certainty score, that $\mathbf{x}$ has true label $y = c$.

$$\text{Cert}(y = c|\mathbf{x}) = \text{Cert}_S(y = c|\mathbf{x})\text{Agr}_G(x) \quad (4)$$

where $\text{Cert}_S(y = c|\mathbf{x})$ is the specific contribution to label certainty, and $\text{Agr}_G(\mathbf{x})$ measures the agreement between models in G about which leaf $\mathbf{x}$ belongs in. $\text{Cert}_S(y = c|\mathbf{x})$ is calculated from the specific extensions $\mathbf{s_k} \in S$ for leaves $l_k$ that select $\mathbf{x}$, and captures both disagreement in the labels of those leaves, and the average weighted overlap of $\mathbf{x}$ of their specific extensions $\mathbf{s_k}$. The individual leaf contributions $\text{Cert}_S(y = c|\mathbf{x})$ are the weighted sum $(\mathbf{w_s} \cdot \mathbf{1_{s_k = x}})$

of literals in each $\mathbf{s_k}$ that also select features in $\mathbf{x}$, normalized by the sum of weights $\|\mathbf{w_s}\|_1$ of all literals in $\mathbf{s_k}$. Then each of these leaf contributions is averaged across the scalar node weights $w_{n_k}$ for each leaf $l_k$ that selects $\mathbf{x}$.

$$\text{Cert}_S(y = c|\mathbf{x}) = \frac{\sum\limits_{n_k : y = c} w_{n_k} \left(\mathbf{w_{s_k}} \cdot \mathbf{1_{s_k = x}}\right) / \|\mathbf{w_{s_k}}\|_1}{\sum_{n_k} w_{n_k}} \quad (5)$$

Calculations for the scalar node weights $w_{n_k} \in [0, 1]$ and extension weight vectors $\mathbf{w_{s_k}} \in [0, 1]^{|\mathbf{s_k}|}$ are described in a later section. Keep in mind that $\text{Cert}(y = c|\mathbf{x})$ is analogous to $P(y = c|\mathbf{x})$, the probability of $y = c$ given $\mathbf{x}$, except that it is not normalized, i.e. usually $\sum_c \text{Cert}(y = c|\mathbf{x}) \leq 1$. It is not normalized because this would cause the uncertainty that $\mathbf{x}$ belongs to one class $c$ to positively contribute to the certainty that it belongs to another. When $\text{Cert}(y = 0|\mathbf{x}) = 0.0$, no model in STAND's $G$ space predicts $y = 0$. However, this doesn't necessarily mean that all parts of $G$ that predict $y = 1$ capture $\mathbf{x}$ in all of their extensions bounded by $S$. In this case, there is still room for uncertainty in whether y=1 (i.e., $\text{Cert}(y = 1|\mathbf{x}) < 1.0$), since STAND may not have narrowed its space of models sufficiently to confidently label $\mathbf{x}$, even if there is no disagreement among the current models' label predictions.

## 3.4. How General and Specific Change with Training

To understand how $G$ and $S$ tend to change in response to new training examples, it is helpful to consider a simplified view of how STAND's data structure changes when a new training example filters into leaves whose majority classes agree with or oppose the true label. When a new training example $\mathbf{x}$ does not satisfy all criteria in a specific extension $s_k$ of a leaf that agrees with its label, it will drop some features, or broaden ranges for continuous valued features, effectively generalizing $S$. In leaves that select $\mathbf{x}$, but disagree with the true label, $G$ will be specialized by turning the leaf into a node with a new set of best feature splits $T_L$. This effectively specializes $G$ by extending all literal chains $t_{L-1}, ..., t_0$ that previously led to that leaf.

For any new training example, the set of highest gain literals $T_L$ for higher nodes can grow or shrink (usually shrink) as more examples are collected. Shrinking $T_L$ for a node shrinks the set of best selection criteria leading to leaves downstream from that node. Consider, for instance, component $G_0$ of $G$ in Figure 1. While a conventional decision tree forms one conjunctive set of selection criteria per leaf (e.g. $\neg X_4 \neg X_3$), STAND compactly represents a set of alternatives:

$$\begin{aligned} \mathcal{G}_0 &= X_4(\neg X_3 | \neg X_5 | \neg X_6) \quad (6) \\ &= X_4 \neg X_3 \mid X_4 \neg X_5 \mid X_4 \neg X_6 \end{aligned}$$

In this notation, the | symbol delimits alternative literals corresponding to edges going into a node. For instance, in

Figure 1, $(\neg X_3|\neg X_5|\neg X_6)$ corresponds to the three edges leading into the left-most node. The full set of literal chains leading to a leaf is the Cartesian product of these options. If new training examples reduce the candidates in any of these sets of options, $G$ will effectively shrink, bringing it closer to enclosing only the true logical statement (or best if the data is noisy). In principle, STAND can achieve something similar to version space convergence where $G = S$ or $\forall_{n_k} G_k = s_k$, when STAND's $G$ reduces to the correct decision tree (which on noiseless data implies perfect holdout set performance), and all extensions $s_k$ in $S$ for each leaf $k$ reduces to the same set of literals as those leading from the root to that leaf (this would additionally cause STAND to ascibe 100% confidence to all holdout set predications).

Training in STAND can be framed as a process of disambiguation that is distinct from typical tree fitting. STAND's systematic treatment of alternative models explicitly captures the ambiguity of selecting the best classifier given limited data, and invites a conceptual shift from the view of model *fitting* as imperfect model selection. A not-deep-enough STAND tree is not helplessly underfit[2] since the elements of $s_k$ capture candidate features that may be part of new nodes in the tree. These candidates allow STAND to reason about the prospective evidence that unseen examples may provide. The $\mathbf{x}$ that minimizes $\mathrm{Cert}_S(y = c|\mathbf{x})$, i.e., violates as many $s_k$ as possible, is the most likely to illustrate that more deepening of G is necessary (or not). A STAND model fit to too little data is also not helplessly underfit in the sense of being an arbitrary best choice among local minima: insofar as there are good alternatives, STAND will capture them, and we can hasten progress toward eliminating these alternatives by selecting examples that maximize the disagreement between them.

### 3.5. General Agreement and Total Label Certainty

Every node in G has a lattice of upstream paths leading into it. A new unlabelled example $\mathbf{x}$ will typically filter down through several paths terminating in one or more leaves. Unlike the previous training examples selected by any node, the new unlabeled $\mathbf{x}$ may only filter through a subset of the paths in all upstream lattices of the nodes that it touches. For instance, several literals may lead into the same node, but $\mathbf{x}$ may only satisfy a subset of them. When trained on $\mathbf{x}$ with a label $y = c$, STAND's G will, in this case, undergo some change; dropping literals in each $T_L$ or restructuring.

The general agreement $\mathrm{Agr}_G(\mathbf{x}) \in [0, 1]$, estimates how

little G needs to change to accomodate $\mathbf{x}$. Let the in-degree opportunity set $O_\mathbf{x}$ be a set of pairs $(t, w_{n_k})$ where each $t$ is a literal gating an ingoing edge of one of the nodes that $\mathbf{x}$ filters through, and each $w_n$ is the node weight of the node $n_k$ upstream of that edge. Each pair $t, w_{n_k} \in O_\mathbf{x}$ represents one opportunity for $\mathbf{x}$ to have filtered through one of the many path segments in the lattices upstream from the leaves it filters into, weighted by the node that each path segment originated from. $\mathrm{Agr}_G(\mathbf{x})$ is the weighted sum of each successful opportunity (i.e., path segment that $\mathbf{x}$ took) divided by the sum of all opportunity weights in $O_\mathbf{x}$:

$$\mathrm{Agr}_G(\mathbf{x}) = \frac{\sum_{t, w_{n_k} \in O_\mathbf{x}} \left( w_{n_k} \cdot \mathbb{1}_{\mathbf{t}(\mathbf{x})} \right)}{\sum_{t, w_{n_k} \in O_\mathbf{x}} w_{n_k}} \tag{7}$$

Summing over path segments instead of full paths makes this calculation simple and non-combinatorial. Additionally, since $\mathrm{Agr}_G(\mathbf{x})$ affects the total probability of each class equally, it does not affect relative label certainties, so if we had normalized $\mathrm{Cert}(y = c|\mathbf{x})$, we would lose the benefits of it. When $\mathrm{Cert}(y = c|\mathbf{x})$ is used for active learning, or interpreting training completion, its contribution down-weights examples that produce high disagreement in $G$, capturing the degree to which learning the label of an unlabelled example may resolve ambiguities about which alternative model in $G$ is the optimal one.

## 4. Hierarchical Shrinkage

Hierarchical shrinkage is a method of post-hoc regularization for tree-based regression that shrinks leaf predictions towards the sample means of their ancestors (Agarwal et al., 2022). For leaf-to-root path $t_L, ..., t_0$, $N(t)$ the number of samples contained at the node for each $t$, and $\hat{\mathbb{E}}_{t_l}\{y\}$ mean node response, hierarchical shrinkage is expressed as:

$$\hat{f}_\lambda(\mathbf{x}) = \hat{\mathbb{E}}_{t_0}\{y\} + \sum_{l=1}^{L} \frac{\hat{\mathbb{E}}_{t_l}\{y\} - \hat{\mathbb{E}}_{t_{l-1}}\{y\}}{1 + \lambda/N(t_{l-1})} \tag{8}$$

Where $\lambda$ is a tunable regression parameter.

In STAND, instead of shrinking leaf predictions,[3] STAND applies hierarchical shrinkage to the label probability estimates that are used to calculate impurity measures like entropy ($E = -\sum_c p_c log(p_c)$) and Gini impurity ($GI = 1 - \sum_c p_c^2$). This is tremendously helpful in low-data ITL situations, where the empirical estimates of each $p_c$ can be based on very little data. This is especially true for deeper nodes that capture tiny subsets of the training data. Consider, for instance, calculating the impurity decrease (the usual gain criterion for decision trees) over different candidate feature splits for a deep node. Within that node's small subset of samples, the positive class $y = 1$ may correlate

---

[2]Note that over-fitting, and its usual remedies (e.g., pruning a too-deep tree), are irrelevant in an ITL pre-condition learning when 100% holdout accuracy is ultimately expected. A less-deep tree with impure leaves is never better than a deeper tree with pure leaves, since impure leaves imply consistencies in training data predictions. Another sense of overfitting is subsample bias in deep nodes, which we address with hierarchical shrinkage.

[3]This almost certainly wouldn't help in typically noiseless precondition induction tasks with a concrete errorless target concept.

highly with certain features, yet not correlate with those features in higher nodes—likely a spurious correlation. To make use of the prior patterns in higher nodes, we apply hierarchical shrinkage, with the regularization parameter $\lambda_p$, to the joint probabilities $P(y = c, X_j = v)$ for each feature selection $X_j = v$ and class $c$. The joint probability correction $\hat{P}^*(c, v)_{t_L}$ at $t_L$ is computed from the empirical joint probabilities $\hat{P}(c, v)_t$ of itself and its ancestor nodes:

$$\hat{P}^*(c, j)_{t_L} = \hat{P}(c, j)_{t_0} + \sum_{l=1}^{L} \frac{\hat{P}(c, j)_{t_l} - \hat{P}(c, j)_{t_{l-1}}}{1 + \lambda_p/N(t_{l-1})} \quad (9)$$

This adjustment has beneficial effects on total model performance, error recurrence rates, and how monotonically STAND's certainty scores $\text{Cert}(y = c|\mathbf{x})$ change on hold-out data throughout incremental training. Similar calculations with shrinkage parameters $\lambda_n$ and $\lambda_s$ are used to calculate the node weights $w_{n_k}$ and specific extension weights $\mathbf{w_s}$ for calculating $\text{Cert}_S(y = c|\mathbf{x})$. Let $\tau_{n_k}$ be the minimum acceptance rate (i.e. $\alpha$) that node $n_k$ requires to remain in $G$. $\tau_{n_k}$ is calculated by looking at all of the the literals $t \in T_{n_k}$ that are upstream of $n_k$ (i.e. that lead into $n_k$), and see how high $\alpha$ needed to be to include it's gain $F_t$ as among the best splits if the maximum gain of splits in that upstream node is $M_t$. The minimum $\alpha$ in each case is $1 - \frac{F_t}{M_t}$, and so $\tau_{n_k} = \min_{t \in T_{n_k}} 1 - \frac{F_t}{M_t}$. From $\tau_{n_k}$ and $N(k)$ the number of examples in node $k$, we calculate $w_{n_k}$ as:

$$w_{n_k} = (1 - \tau_{n_k})/(1 + \lambda_n/N(k))) \quad (10)$$

The denominator of $w_{n_k}$ downweights nodes with lower example counts $N(k)$. The numerator $(1 - \tau_{n_k})$ downweights nodes with literals in their upstream lattice that are only present in $G$ because the acceptance rate $\alpha$ permits literals in their upstream lattice with less than maximal gain.

The calculation of $\mathbf{w_{s_k}}$ on the otherhand applies hierarchical shrinkage to estimate the prior probability $\hat{\mathbf{P}}^*(s_k)$ that each element in $s_k$ is invariant among all examples of class $c$, by shrinking over the empirical conditional probabilities $\hat{P}(X_t^{s_k}|c)$ that samples share $s_k$'s features when $y = c$ in each ancestor node.

$$\hat{\mathbf{P}}^*(s_k|c) = \hat{\mathbf{P}}(X_{t_0}^{s_k}|c) + \sum_{l=1}^{L} \frac{\hat{\mathbf{P}}(X_{t_l}^{s_k}|c) - \hat{\mathbf{P}}(X_{t_{l-1}}^{s_k}|c)}{1 + \lambda_s/N(t_{l-1})} \quad (11)$$

Each element of $\mathbf{w_{s_k}}$ is the posterior probability that each feature in $s_k$ is invariant. To calculate this posterior probability we adjust a Jeffrey's prior of $Beta(\frac{1}{2}, \frac{1}{2})$ by adding in $\hat{\mathbf{P}}^*(s_k|c)$, as $Beta\left(\frac{1}{2} + \hat{\mathbf{P}}^*(s_k|c), \frac{1}{2} + 1 - \hat{\mathbf{P}}^*(s_k|c)\right)$.

$$\mathbf{w_{s_k}} = \frac{\left(N(k) + \frac{1}{2} + \hat{\mathbf{P}}^*(s_k|c)\right)}{N(k) + 2} \quad (12)$$

## 5. Methods

We compare STAND with and without hierarchical shrinkage with parameters $\lambda_p = 25$, $\lambda_s = 25$, $\lambda_n = 50$ (tuned in Appendix D) to several alternative models:

1. **Decision Tree**: CART decision trees using gini impurity (Breiman et al., 2017) as the impurity criterion. We also train a probability shrinkage condition with $\lambda_p = 25$, ($\lambda_n$ and $\lambda_s$ are specific to STAND)

2. **Random Forest**: Ensemble (Breiman, 2001) of 100 decision trees. Random forests use bagging (Breiman, 1996) to train several decision trees on subsets of data.

3. **XG Boost**: Gradient boosting-based ensemble of decision trees. Trees are trained sequentially over the errors of the previous ones (Chen & Guestrin, 2016).

4. **VSSM**: An incremental version space method that can learn disjunctive concepts by splitting and merging separate version spaces over conjunctive concepts (Hong & Tseng, 1999).

5. **Neural Network**: Fully connected with 3x100 (ReLU) hidden layers, Adam optimizer (lrn. rate=1e-3).

VSSM and the tree-based methods excel at learning from small datasets of structured data. For tree models, we set no limits on tree depth or leaf size, which can only harm performance in noiseless precondition learning tasks. For the remaining tunable parameters, a simple grid search found no benefits to deviating from the scikit-learn defaults. All models are evaluated on holdout set performance and error reoccurrence rates as training examples are collected. We compare STAND with the two ensemble methods and the neural network in terms of how their prediction probabilities on holdout data evolve through training. This includes how monotonically predictions increase toward the correct prediction, how closely predictions align with true holdout set precision, and how helpful they are for active learning selection within the training set.

### 5.1. Precondition Learning Tasks

With VAL, we evaluate precondition learning in Dice Adventure, a multiplayer strategy game (Zhang et al., 2025), and with AI2T, we evaluate on a multi-column addition and fraction arithmetic tutoring system (see Appendix C). We report performance over 40 repetitions each, where an oracle grades candidate rule applications as the agent takes actions. In these tasks, the agents' core learning mechanisms rapidly induce base rules that can take actions immediately from the first examples. However, these induced base rules lack preconditions to control their correct application. We evaluate STAND and the comparison models at the slightly slower precondition induction phase of training, driven by users' grading several performance attempts.

*Table 1.* Synthetic data final accuracy %, and average false positive (FP) and false negative (FN) error reoccurrence rates % per fit on holdout examples (± std. error $\sigma/\sqrt{n}$).

| Model | Accuracy | FP Reocc. | FN Reocc. |
|---|---|---|---|
| STAND | $90.0 \pm 1.1$ | $2.6 \pm 0.1$ | $1.2 \pm 0.1$ |
| STAND (h.s.) | $\mathbf{92.8 \pm 1.1}$ | $\mathbf{2.2 \pm 0.1}$ | $0.8 \pm 0.1$ |
| Dec. Tree | $88.6 \pm 1.1$ | $7.3 \pm 0.3$ | $3.8 \pm 0.2$ |
| Dec. Tree (h.s.) | $87.81 \pm 0.9$ | $8.9 \pm 0.3$ | $4.3 \pm 0.2$ |
| XGBoost | $84.8 \pm 1.5$ | $3.4 \pm 0.1$ | $0.8 \pm 0.1$ |
| Rand. Forest | $56.2 \pm 0.8$ | $2.4 \pm 0.2$ | $\mathbf{0.2 \pm 0.1}$ |
| Neural Net | $56.7 \pm 0.7$ | $3.3 \pm 0.1$ | $1.7 \pm 0.1$ |
| VSSM | $55.5 \pm 1.5$ | $0.04 \pm 0$ | $0.03 \pm 0$ |

To avoid ceiling effects between models, we also train on a synthetic dataset that simulates a challenging ITL precondition learning task over 100 training examples with 400 features and 2000 holdout set examples. We report performance over 100 repetitions. In the synthetic data, target preconditions are disjunctions (i.e., ORs) of two conjunctive concepts with $1 + \text{Poisson}(\lambda=1)$ random non-overlapping literals. Models are trained one example at a time over the sequence of 100, and negative examples taper off later in the sequence. Each sequence of 100 is imbalanced, with only 20 negative examples in total. Features are strictly categorical (which is common in these tasks) and partially structured by increasing the rate at which feature values tend to co-occur, making spurious co-occurrences between labels and certain feature values common and persistent. The data is also explicitly treated to prevent partial discovery of the target concepts, such as obtaining 2 of 3 literals in a conjunction, which is insufficient to perform well in the holdout set (see Appendix B for further details).

## 6. Results

Across evaluation tasks, we report several desirable measures of effective, monotonic inductive learning driven by a human-in-the-loop instructor in an ITL setting. We report each result in terms of incremental improvement as each example is collected during simulated interactive training (see Appendix D for extended non-synthetic tasks results).

### 6.1. Holdoutset Performance by Example

We find that across all precondition learning tasks, STAND matches or has higher holdout set performance than the comparison models (see Figure 2). In the synthetic data (see Figure 2 & Table 1) after N=100 examples, STAND (90.0%) outperforms decision trees (88.6%) and considerably outperforms XGBoost (84.8%). At N=100 examples, STAND performs even better with hierarchical shrinkage (92.8%), but shrinkage ($\lambda_p$=25.0) did not aid decision trees (87.81%). VSSM (55.5%), random forests (56.2%), and neural networks (56.7%) perform poorly at these tasks. VSSM is likely ill-suited to the large feature space, while

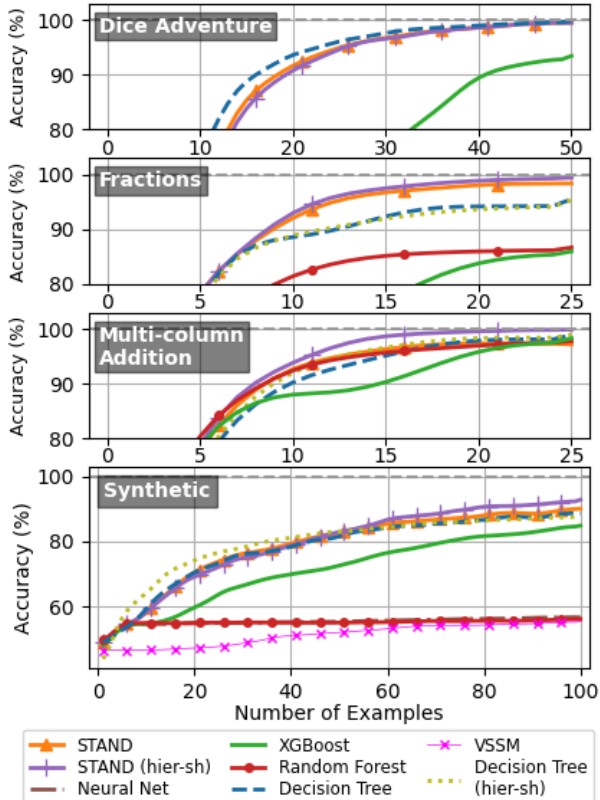

*Figure 2.* Each model's holdout accuracy by example for Dice Adventure (via VAL), Fractions and Multi-column Addition (via AI2T), and synthetic data. STAND with hierarchical shrinkage matches or exceeds other models.

random forests and neural networks lack the sample efficiency to succeed at these tasks[4]. On the non-synthetic tasks grounded in ITL systems, there is an appreciable gap between STAND's nearly 100% final accuracy with hierarchical shrinkage (Dice Adventure: 99.51%, Fractions: 99.88%, MC Addition: 99.43%), and the next best tree model, the Decision Tree (Fractions: 98.8%, MC Addition: 95.4%). In 39/40 repetitions for dice adventure, 35/40 repetitions for Fractions, and 16/40 repetitions for MC addition, STAND with hierarchical shrinkage achieved 100% holdout accuracy.

### 6.2. Error Reoccurrence

In an ITL setting with a human instructor, it is helpful for the model's behavior to change consistently as it is trained—strictly eliminating errors without them recurring upon refitting. As we discussed in the introduction, instances of false positive reoccurrence are particularly troublesome and can lead to missed opportunities for negative feedback. Excluding VSSM (which has impractically low accuracy), in the synthetic data, STAND with hierarchical shrinkage has the lowest rate of false positive reoccurrence (2.2%), and a low

---

[4]Forest subsamples probably exclude important edge cases

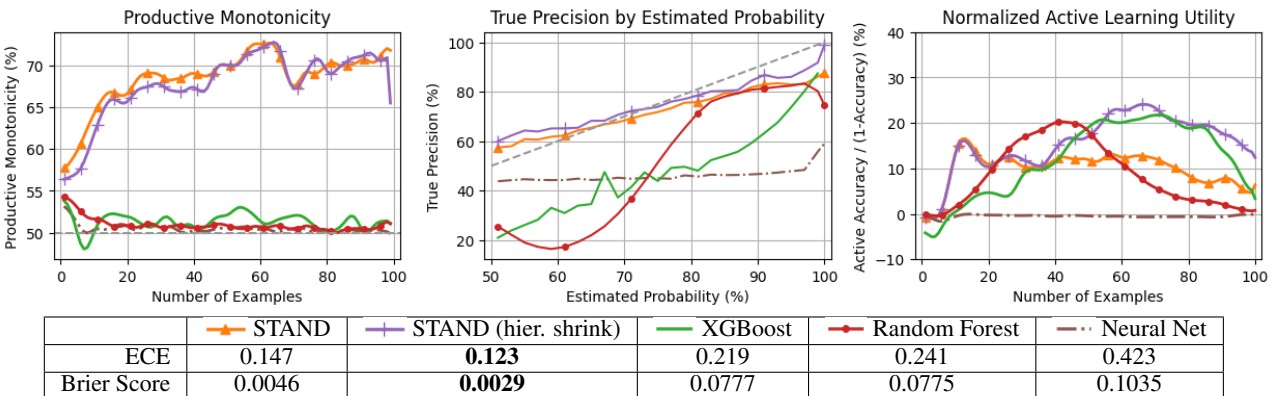

| | STAND | STAND (hier. shrink) | XGBoost | Random Forest | Neural Net |
|---|---|---|---|---|---|
| ECE | 0.147 | **0.123** | 0.219 | 0.241 | 0.423 |
| Brier Score | 0.0046 | **0.0029** | 0.0777 | 0.0775 | 0.1035 |

*Figure 3.* For synthetic data, graphs from left to right: the productive monotonicity by number of examples, true precision by estimated probability, and normalized active learning utility by example. The table below the graphs shows the Expected Calibration Error (ECE) and Brier Score (lower is better) over holdout precision in the center graph. Only prediction probability capable models are shown.

(but not best) false negative reoccurrence rate (0.8%). False negative errors correspond to rules not being applied when they should be. In an ITL scenario, the human instructor experiences this as an agent impasse, which is easily resolved by providing a worked example.

### 6.3. Productive Monotonicity

A model's ability to predict its probability of correctness on unlabelled examples can be an essential tool for instructors in an ITL setting to estimate the learning progress of the inductive systems they are training. In rule precondition learning, this can take the form of actions proposed with a certainty percentage, or full rollouts of planned behavior marked with action-by-action certainty. Ideally, these measures of certainty should increase toward 100% as performance on holdout data improves. We define **productive monotonicity** as the proportion of *prediction probability changes* on holdout examples (only changes $> 2\%$, to filter small jitters), where the change increases the probability (or STAND's $\text{Cert}(y = c|\mathbf{x})$) toward the correct label.

Figure 3 (left) shows that STAND is considerably better than the comparison models in terms of productive monotonicity. The alternatives show rates that are not much higher than 50%, while STAND has rates of 60-70% after about 10 examples. This highlights STAND's remarkable property of being a self-aware learner. STAND can actually estimate when new examples produce meaningful learning.

### 6.4. Absolute Precision Calibration

Another element of STAND's self-awareness is that its prediction probabilities align very closely with actual prediction precision on holdout data. We measure precision here (not accuracy) since our ITL systems only show users' actions where $y = 1$ is the prevailing prediction. Figure 3 (middle) plots the precision of each model's probability estimates

(or $\text{Cert}(y = c|\mathbf{x})$) aggregated into 25 bins (width $\pm 1\%$). Expected Calibration Error (ECE) and Brier Score measure the difference between each model and a perfect one-to-one alignment of true holdout precision with estimated probability. STAND with hierarchical shrinkage has by far the lowest ECE (0.123) and Brier Score (0.0029), and most importantly, its estimates of 100% probability have a nearly perfect precision of 99.71%. Other models show very poor precision calibration, with the neural network's predictions showing almost no adherence to the true precision rates. These results illustrate that STAND can estimate how much its performance improves on unlabelled holdout scenarios, and when its learning terminates at 100% holdout accuracy.

### 6.5. Active Learning Utility

Figure 3 (right) plots active learning utility: the models' accuracy as an active learner (Active Accuracy) normalized by the average error (1-Accuracy) in regular training. When trained with active learning, each model predicts the label probabilities of examples in a training pool (twice the size of the usual training set), and selects the most uncertain example instead of a random one. STAND with hierarchical shrinkage has the best active learning utility in the first 20 problems, with similar utility as XGBoost thereafter. However, since STAND has a higher absolute baseline performance, it is arguably more effective in these regions as well.

### 6.6. Recognizing User Errors

In addition to selecting helpful next training examples, STAND's certainty scores can be used to identify candidate training examples that users may have mislabelled as correct or incorrect. Table 2 shows models trained on the synthetic data with one training example label flipped. Model predictions on training examples are evaluated by using leave-one-

*Table 2.* Mislabeled sample detection: percentage of trials where the flipped sample ranks in the top-1 or top-5 most uncertain predictions under leave-one-out retraining.

|  | #Train = 20 | | #Train = 50 | |
| --- | --- | --- | --- | --- |
| Model | Top-1 | Top-5 | Top-1 | Top-5 |
| RandomForest | 29.1% | **86.1%** | 35.2% | 65.3% |
| XGBoost | 18.0% | 79.2% | 52.2% | 75.8% |
| STAND | 36.4% | 75.9% | 51.2% | **80.4%** |
| STAND (h.s.) | **37.8%** | 80.5% | **55.9%** | 79.1% |

out retraining to predict each training example[5]. Table 2 shows that for 50 training examples, STAND with hierarchical shrinkage picks the flipped example as the least certain example 55.9% of the time in 1000 repetitions (comparable to XGBoost at 52.2%), but when there are only 20 examples, it is much better than the ensembles (37.8% versus 18% for XGBoost and 29.1% for RandomForest). In both scenarios, there is about an 80% chance that the flipped example is in the top 5 most uncertain examples. These results show that STAND has the potential to help users identify their mistakes. For instance, in an ITL setting, users could be shown a list of the top 5 most anomalous training examples to guide their attention for locating mistakes and identifying cases that are poorly supported by their training so far.

### 6.7. Scalability and Robustness to Noise

Many big-data machine learning problems are inherently larger and noisier than the ITL use cases we focus on here, rendering 100%-accurate predictors infeasible. Evidently, STAND is competitive (but not decidedly state-of-the-art) in these scenarios as well. Appendix E includes a comparison on 6 public UCI classification datasets[6]: *breast-cancer*, *hepatitis*, *soybean*, *tic-tac-toe*, *vote*, and *zoo*. STAND is comparable to other tree models in these cases, with a mean test accuracy of (89.18%), slightly better than a CART-based Decision Tree (89.12%), but marginally worse than the Random Forest (89.67%), and XGBoost (89.51%). These results show that despite borrowing ideas from version space learning, STAND does not collapse under noise like conventional version spaces. Additionally, we found that on very large and noisy datasets STAND's node-caching trick maintains a manageable size, requiring only about twice as many nodes as a conventional decision tree—far from the millions of nodes needed in a full option tree. Considering that STAND is faster to train than typical finite ensembles, but more exhaustive in terms of the space of models it cap-

---

[5]Leave-one-out retraining is necessary, since STAND will always report 100% confidence for examples in its training set (this is still quite fast, typically less than 1 second for 50 examples, versus 7-23 seconds for RandomForest and XGBoost)

[6]https://archive.ics.uci.edu/

tures, we believe there is future potential for adaptations of STAND that are specialized for tasks beyond small-data ITL precondition learning.

## 7. Discussion

STAND's self-aware learning succeeds along dimensions that regularly stump modern data-driven machine learning approaches. Deep learning, for instance, typically fails to learn from limited data and rarely achieves 100% accuracy, in part because of how gradient descent encodes capabilities, and in part because of the inevitable errors in the large-scale data needed to train it. The popularity of these big-data-oriented approaches has entrenched an expectation, as almost a rule, that machine learning systems are fundamentally imperfect. At a tantalizing $> 99.4\%$ accuracy on real ITL precondition learning tasks, STAND is not perfect either, but it does provide uniquely powerful information for user-driven improvement by accurately estimating its own learning progression. Future work may explore how this ITL-oriented approach—in which a user interactively trains AI agents with robust, reproducible, and interpretable capabilities—might augment or replace generative AI in long-horizon step-by-step tasks like web and operating system automation. The state-of-the-art in generative AI for these tasks hangs at an abysmal $\sim 30\%$ task completion rate (OpenAI, 2025; Ma et al., 2024; Manus, 2024). By contrast, within the ITL systems available to us, and the kinds of tasks they have been designed for, our ITL learning approach with STAND is quite robust.

## 8. Conclusion

We have introduced STAND and illustrated its data-efficient, self-aware learning. In an ITL setting, these qualities can help support effective human-in-the-loop instruction. STAND borrows qualities from version space learning, such as representing a space of generalizations with two boundary sets, yet eliminates the fragility and representation limitations of candidate elimination-based version space learning. A key theme of STAND's approach is a systematic and efficient characterization of all good alternative models, enabling an explicit treatment of the ambiguity of converging on an optimal model from limited data. By modeling this ambiguity explicitly, STAND can self-assess its learning progress by estimating remaining disagreement among its space of consistent models, which we show can be significantly more effective than finite ensemble approaches in estimating performance on unseen examples. We suspect that other applications, beyond precondition induction, may also benefit from a similar approach, especially when applied to human-in-the-loop training, where model self-awareness is particularly helpful.

## Impact Statement

This work supports a broad vision of interactive task learning (ITL) systems that can be trained directly from natural instruction. The impact of realizing this vision is far-reaching and can be distinguished from the impact of data-driven machine learning along dimensions of expected accuracy, trust, privacy, and potential for private ownership. This work is particularly concerned with learning interpretable knowledge structures that automate routine and concretely definable tasks. And thus may contribute to aligning learned AI capabilities with the expectations of conventional computer applications, whereby a program serves a well-defined purpose without error. Many have complained that the public-facing AI systems of the 2020s have contradicted their expectations: they can generate art and write poems, but cannot be trusted to do our taxes. Despite considerable investment, to date, few vendors of large generative models have succeeded at making a profit from them. This is perhaps because tasks of substantial economic value tend to hinge on reliability, and the utilization of skills and information that cannot be easily scraped from the internet.

Machine learning systems that learn interactively and robustly from human expertise may reliably replace human labor, even for bespoke tasks for which it would be impractical to collect large training datasets. Moreover, the trainers of those AI agents could take personal ownership (maintaining privacy and intellectual property) and hold personal responsibility (maintaining legal liability) for their programmed behavior and individual actions, especially if the learned knowledge can be interpreted and audited. Computationally efficient and data-efficient learning algorithms like STAND also require very little power for learning and inference, meaning they have a low environmental impact and contribute very little to the spikes in hardware and energy prices caused by the current AI boom. Interactively teachable AI also offers broad possibilities in education, enabling a teacher to tutor an AI agent to generate an expert system that can tutor students directly at scale (e.g., like AI2T (Weitekamp et al., 2024)) and to be used as a tool for cognitive analysis of domain content. Moreover, such agents offer the possibility of simulating student learning (not just behavior) as a scientific tool for advancing the learning sciences.

## Acknolwedgements

The material in this paper is based in part upon work supported by the National Science Foundation under Grant No. 2247790 and Grant No. 2112532. Any opinions, findings, and conclusions or recommendations expressed in this material are those of the author(s) and do not necessarily reflect the views of the National Science Foundation

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

# A. Appendix A: Additional Notes on STAND

### A.1. Dynamic Acceptance Rate $\alpha_{n_k}$

Our STAND implementation varies the acceptance rate $\alpha_{n_k}$ for each node $n_k$ as a function of the number of training samples $N(k)$ they select, as follows:

$$\alpha_{n_k} = 1 - \min((1 - \alpha_0) + \alpha_0(N(k)/M), 1) \quad (13)$$

Where the tunable parameter $\alpha_0 = .1$ is the base acceptance rate, and the parameter $M = 50$ is the number of samples where a node's $\alpha$ is reduced to $0.0$.

The intuition behind having the acceptance rate $\alpha$ decrease with respect to $N(k)$ is that when we have few samples, we want the set of literals $T_L$ expanded at each node to be large so that we are not excluding any potential optimal literal $t^*$, which could potentially produce the highest gain if we had collected sufficient training data. As more examples are collected, our estimate of the gain of each $t \in T_L$ becomes more precise, so we can lower the acceptance threshold and narrow down the set of literals that are still considered.

By reducing the size of $T_L$, especially in higher nodes and later in training, this approach also limits how many unique subtrees are captured in STAND's $G$ set, which limits the chance of needing to fit large trees.

### A.2. Fit Time and Pseudo-Incremental Fitting

STAND's node caching trick makes it remarkably fast at refitting despite capturing a large space of possible classifiers. We find that STAND can be fully refit in just slightly more time than a regular decision tree (see Figure 5, and considerably less time than ensemble methods like XGBoost and random forests.

Additionally, STAND can be fit pseudo-incrementally. STAND does not randomly break ties when selecting splits, so its structure after refitting on a new example often looks very similar to its structure before the refit. Typically, the new example will reduce $T_L$ sets in nodes or deepen leaves, and much of the previous structure of the STAND tree can be reused. Incremental fitting in STAND only re-fits newly created subtrees, and otherwise updates feature counts and other statistics as needed. In our experiments, this fitting approach consistently shows sub-millisecond fit times for each new example.

### A.3. Sequential Convering with STAND

The sequential covering variant of STAND operates largely the same as the divide-and-conquer approach, with a few crucial differences. Sequential covering approaches, such as FOIL (Quinlan, 1996), try to greedily build several con-

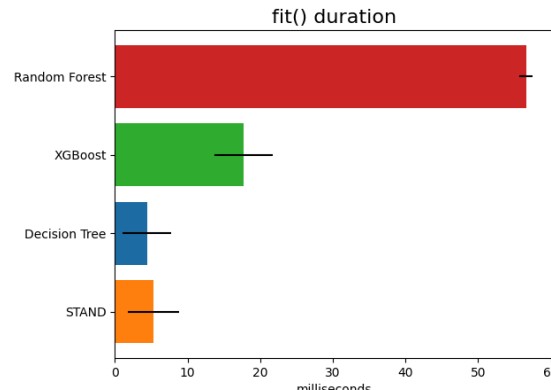

*Figure 4.* Average fit times in milliseconds for random forest (100 trees), XGBoost, decision tree, and STAND on fractions task. Neural net and VSSM are not shown (to maintain scale) since they take several seconds, which is not ideal for interactive training.

juncts one literal at a time that select only positive examples while rejecting all negative examples. Each time a conjunct is found that selects only positive examples, a new conjunct is constructed to try to capture the remaining positive examples and reject all the negative ones. The path of selection criteria through each conjunct is analogous to a root-to-leaf path in a tree terminating in a positive leaf.

As in the divide-and-conquer variant of STAND, these paths compactly capture several alternatives in a lattice. Yet, since there is no recursive splitting of the samples that do not satisfy each literal, negative examples cannot filter into alternative leaves. There is only one negative leaf, which selects all negative examples. In our synthetic dataset, we find that the sequential cover variant of stand is not quite as effective as the divide-and-conquer variety.

### A.4. Relative Featurization

In the non-synthetic domains (Dice Adventure, Fractions, and Mult-column addition), STAND is used in conjunction with an ITL system (VAL or AI2T). Other learning mechanisms within these systems learn hierarchical rules (i.e., HTN methods) from demonstration and/or natural language. STAND's role is to amend these rules with sets of preconditions that gate their correct execution. A training example for STAND in this setting is a candidate application of a rule, situated within a particular environment state.

In many cases, the individual features within the environment are not helpful for precondition learning in their grounded form. Rule preconditions may need to be expressed in terms of relational predicates that generalize across different candidate applications of the same rule. A simple approach is to swap any grounded values in feature predicates with the variable they are bound to in each candidate rule application. However, this simple approach

can only re-express a small number of feature predicates that overlap with arguments in each rule.

Instead, we use the approach described by Weitekamp et al. (2025), which they term relative featurization, where the entire state is restructured with respect to the arguments of candidate rules by chaining together available grounded spatial relations, such as ('left_of', 'thing1', 'thing2'). Each feature is restated using the shortest path from each rule argument to that feature. Shortest paths are found using the Floyd-Warshall algorithm (Floyd, 1962) over the whole state.

# B. Appendix B: Synthetic Data Preparation

Our synthetic data simulates challenging precondition induction scenarios with feature properties similar to our real ITL tasks, yet more difficult in terms of the target generalization to avoid ceiling effects among models.

First, a feature matrix $X$ with 2100 samples and 400 integer categorical features is generated. The number of categorical values per feature is sampled from $2 + Poisson(1)$.

Next, a set of target preconditions is generated. Each precondition set is a disjunctive logical statement with two conjuncts, each with $1 + Possion(1)$ non-overlapping literals. The feature value $v$ for each literal $X_j = v$ is sampled uniformly among the possibilities for each feature.

So far, the feature matrix $X$ is unstructured, and features are mostly uncorrelated. Real ITL environments typically have fairly structured states that change incrementally between timesteps. Thus, these features co-occur often. To simulate random patterns that co-occur in the data, we sample 100 conjuncts with $2 + Possion(3)$ literals. Literals in these conjuncts overlap with previously sampled literals 20% of the time. For each sample in $X$, each of the 100 conjuncts has an 80% chance of being applied to the sample (i.e., the conjunct's values become the feature's values).

Next, each target conjunct is applied to 28% of the samples in $X$, which jointly results in at least one being applied to about half of the data on average. Samples in $X$ that already satisfy either of the conjuncts count toward these totals. Each of these applications of the target conjuncts is given a value of 1 in the label vector $Y$.

Finally, for each sample $X_j$ with label $Y_j \neq 1$, we apply both target conjuncts, and then with 10% probability per literal in each conjunct, resample the feature's value. At least one feature per sample is always resampled so that all conjuncts are no longer satisfied. Many samples will be in a state where they are nearly satisfied. This final step ensures that it is not possible to perform well on the holdout set by learning a subset of each target conjunct, and that there is some variability in conjuncts that are violated in the negative training examples.

The 2100 samples are randomly split into 100 training examples and 2000 test examples for evaluation. The 100 training examples are selected so that there are only 20 negative examples, and those negative examples are more likely to occur earlier in the training sequence. This simulates the fact that as agents learn, they tend to perform tasks correctly, producing more positive examples over time.

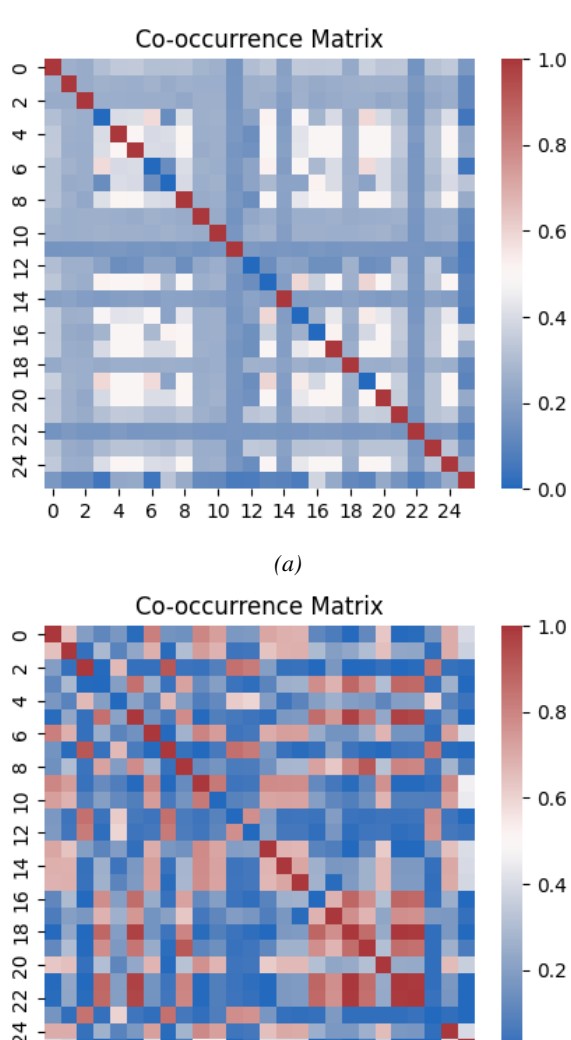

*(a)*

*(b)*

*Figure 5.* Co-occurrence rate $P(X_i = v \land X_j = v)$ of categorical feature values for pairs of features $X_i$ and $X_j$. Matrix (a) shows co-occurrence just from uniform sampling. Matrix (b) has been structured by applying several conjuncts to the data at a rate of 80%.

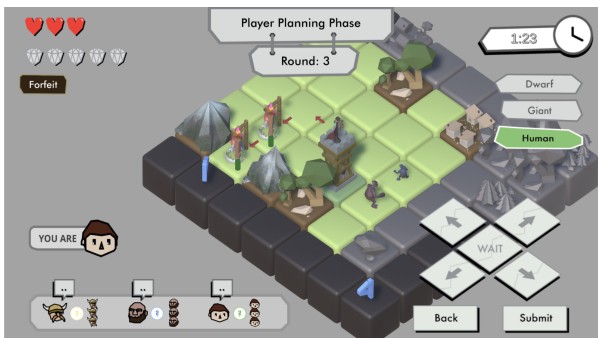

*Figure 6.* Screenshot of Dice Adventure.

# C. Appendix C: ITL Domains

## C.1. Dice Adventure

Dice Adventure (Zhang et al., 2025) is a three-player cooperative strategy game, where each player takes on the role of the human, the dwarf, or the giant. Dice Adventure was designed to investigate human-AI teaming with asymmetric roles. Players must locate their character's orb on a partially observable grid map filled with obstacles and enemies. After retrieving their orb, they must return it to the tower. Each character differs in their sight range, energy, attack power, and ability to destroy obstacles like rocks. The game cycles between a pinning phase, where characters communicate with no-descript markers, and a planning phase, where they register a sequence of acts to be carried out on their turn.

We begin with a set of complete HTN methods sufficient for each character to beat the game. We pre-load VAL with those methods with their pre-conditions removed and learn the pre-conditions inductively by using the original HTNs as an oracle grader.

## C.2. Multi-Column Addition

A tutoring system that teaches the algorithm for long addition over pairs of 3-digit by 3-digit numbers. Partial sums are solved from right to left. The ones digit is placed below, and if the partial sum is 10 or greater, the agent must carry the 1. AI2T (Weitekamp et al., 2024) agents are trained on this domain interactively using TutorGym (Weitekamp et al., 2025b). The agents directly learn HTN methods from demonstrations, and the preconditions are induced from grading correct and incorrect applications of those methods.

## C.3. Fractions

A tutoring system that teaches three different fraction arithmetic procedures: adding fractions with the same denominator, adding fractions with different denominators, and multiplying fractions. As with Multi-column addition, AI2T agents are taught interactively through TutorGym.

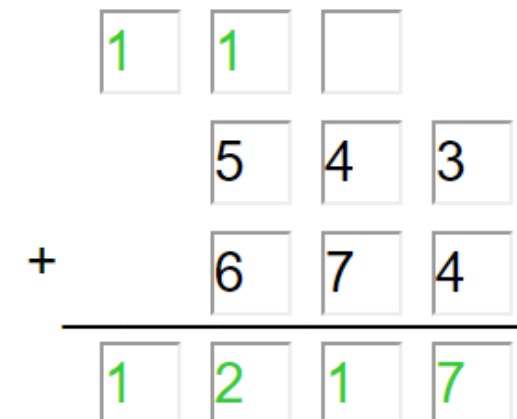

*Figure 7.* Multi-column addition domain.

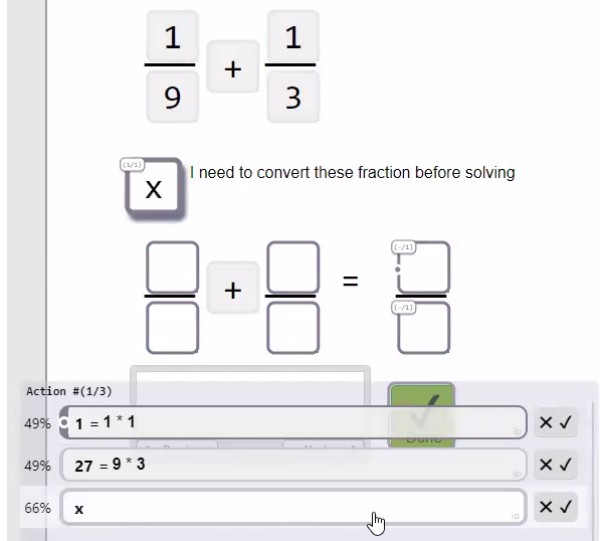

*Figure 8.* Fraction arithmetic domain (Shown in the interactive AI2T interface).

# D. Appendix D: Extended results and Hyperparameter Search Domains

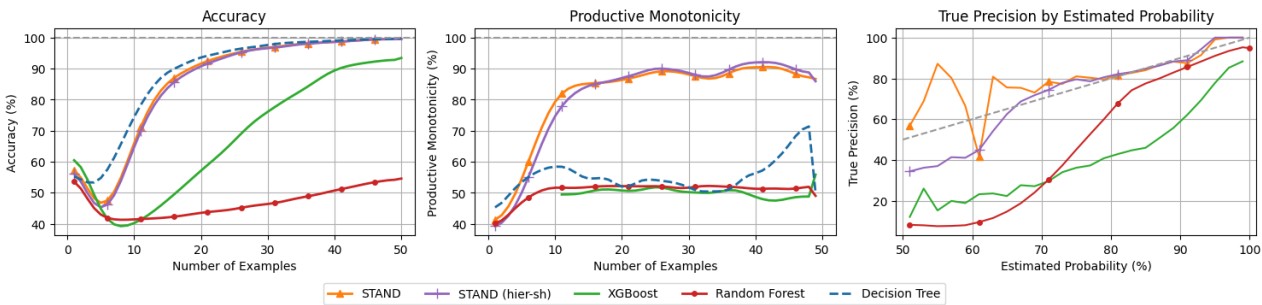

*Figure 9.* Dice Adventure: Accuracy, Productive Monotonicity, and True Precision by Estimated Probability

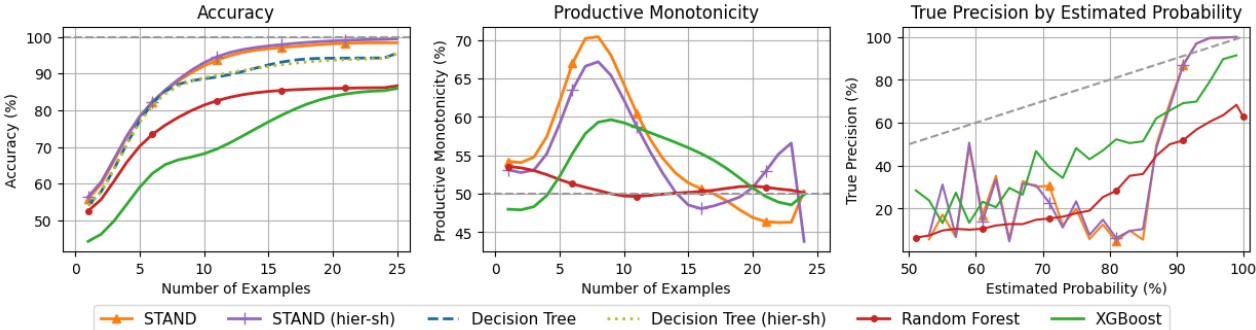

*Figure 10.* Multi-column Addition: Accuracy, Productive Monotonicity, and True Precision by Estimated Probability

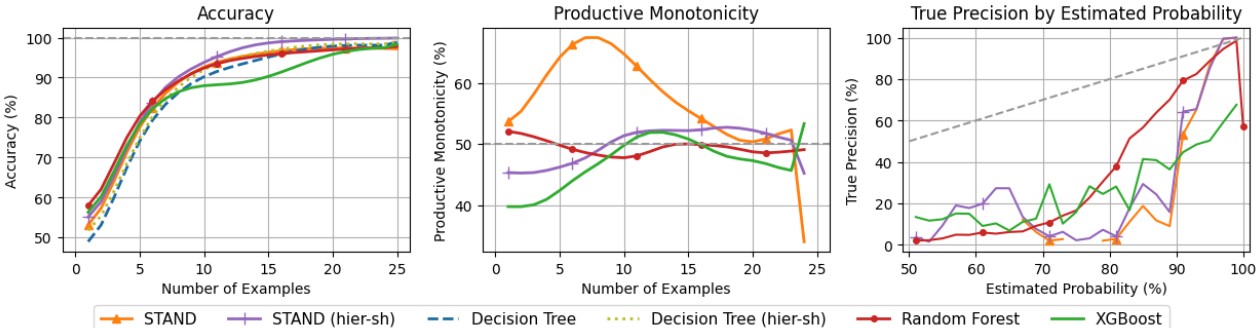

*Figure 11.* Fraction Arith: Accuracy, Productive Monotonicity, and True Precision by Estimated Probability

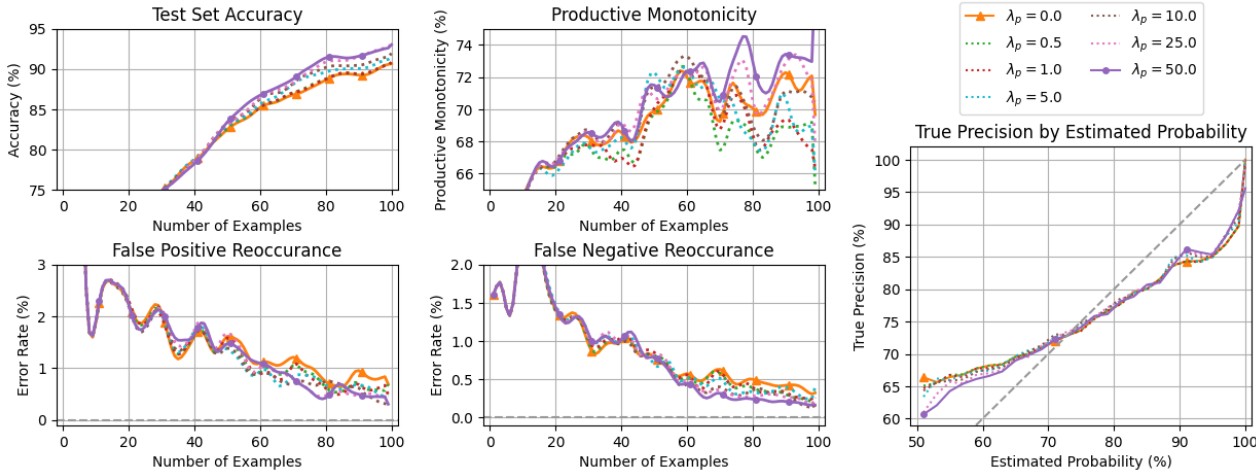

*Figure 12.* Hyperparameter search of $\lambda_p$ over $\{0, 0.5, 1.0, 5.0, 10.0, 20.0, 50.0\}$ higher $\lambda_p$ can benefit accuracy, productive monotonicity, and error reoccurrence. The choice of $\lambda_p = 25.0$ retains these benefits while maintaining high precision in the neighborhood of prediction probabilities close to 100%.

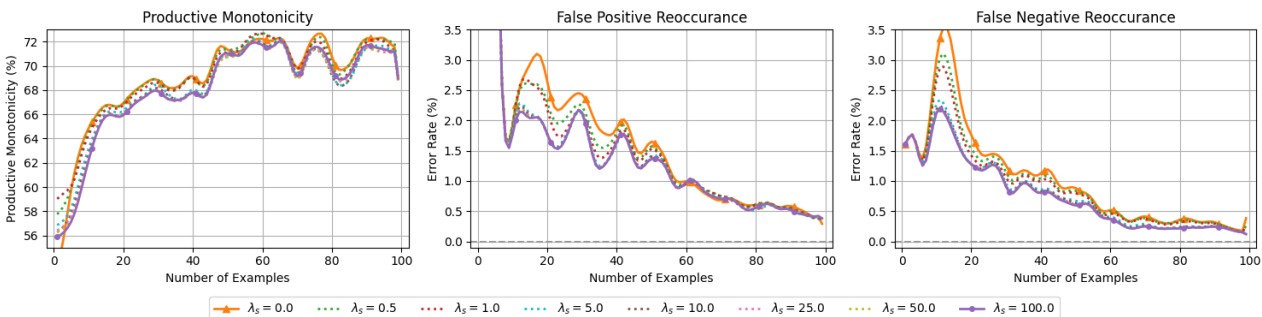

*Figure 13.* Hyperparameter search of $\lambda_s$ over $\{0, 0.5, 1.0, 5.0, 10.0, 20.0, 50.0, 100.0\}$ higher $\lambda_s$ has a minor negative effect on productive monotonicity but reduces error reoccurrence. The choice of $\lambda_s = 25.0$ achieves the maximum reduction in error reoccurrence.

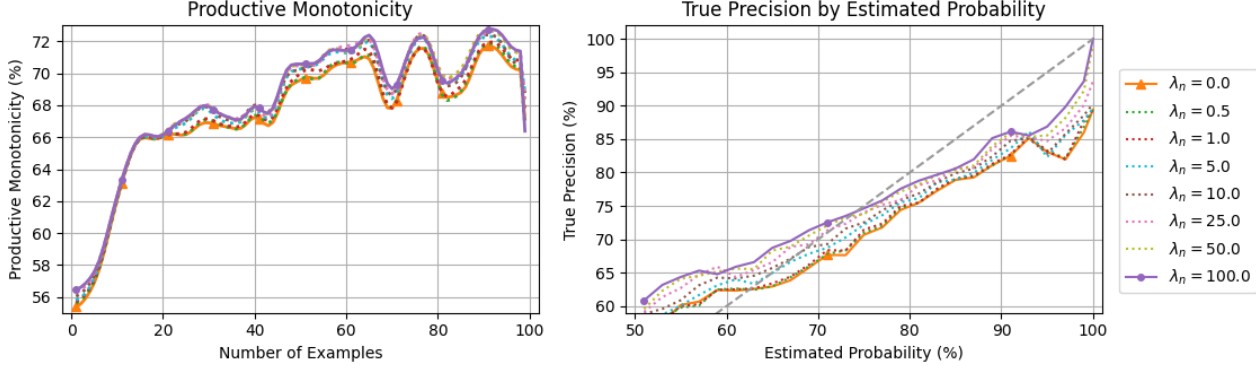

*Figure 14.* Hyperparameter search of $\lambda_n$ over $\{0, 0.5, 1.0, 5.0, 10.0, 20.0, 50.0, 100.0\}$ higher $\lambda_n$ has a minor positive effect on productive monotonicity and improves precision at high prediction probabilities. The choices of $\lambda_n = 50.0$ and $\lambda_n = 100.0$ similarly improve productive monotonicity and absolute precision rates.

# E. Appendix E: Evaluation on Noisy Datasets

This appendix reports a comparison of STAND to the other tree models we used above: DecisionTree, Random Forest, and XGBoost, in a conventional data-driven classifier training task, where models are fit to a fixed set of training data and evaluated on a separate test dataset. This evaluation shows that STAND is comparable to other tree classifiers in noisier, larger datasets than those in our target ITL tasks. These tasks differ considerably from the precondition learning tasks that make up the main body of this work in a few important respects:

1. These datasets consist of hundreds of examples rather than only tens.

2. They are not generated from an interactive training process; thus, there is no selective pressure limiting the frequency of particular kinds of examples. For instance, in interactive precondition learning systems, negative examples tend to be infrequent, attenuate over time, and be dissimilar since the system tends not make similar mistakes twice.

3. These datasets are also noisy, in the sense that they are not perfectly separable (with the exception of tic-tac-toe). There is no perfect class predictor, like in rule precondition learning, that can separate signal from noise.

For each dataset, we randomly sample 100 independent 90%/10% train/test splits. The Random Forest is instantiated with N=100 trees. To avoid overfitting, a max node depth is imposed on all models for each dataset, shown in Table 3 (these choices were fine-tuned for the Decision Tree classifier). STAND with hierarchical shrinkage uses the same parameters as above ($\lambda_p = 25.0$, $\lambda_s = 25.0$, $\lambda_n = 100.0$). The results, reported in Table 4 (next page), show that STAND is comparable to other tree models when trained on large, noisy data, and does not collapse under noise like a classic version space.

Table 5 (next page) illustrates that even in very large and noisy datasets (e.g., *satimage*, *letter*, *isolet*), STAND's node caching trick keeps its lattice at a manageable size. Even when configured with a very permissive acceptance rate of $\alpha = 0.2$ and no depth limit, STAND usually only needs about twice the nodes required by a decision tree. Compared to the millions of nodes needed to construct the equivalent option tree, STAND's node-caching approach, in which nodes are combined if they select the same training examples, appears to be an effective way to capture an exhaustive set of good models with a very low memory footprint. While our results do not show that STAND performs decidedly better than the comparison models in big data scenarios, we

*Table 3.* Dataset Information

| | Dataset (UCI #) | Size | #Feat. | Max Depth |
|---|---|---|---|---|
| large | breast-cancer (14) | 286 | 9 | 2 |
| | hepatitis (155) | 155 | 19 | 4 |
| | soybean (90) | 683 | 35 | 20 |
| | tic-tac-toe (101) | 958 | 9 | 20 |
| | vote (105) | 435 | 16 | 2 |
| | zoo (111) | 101 | 16 | 8 |
| very large | satimage (146) | 6430 | 36 | – |
| | letter (59) | 20000 | 16 | – |
| | isolet (54) | 7797 | 617 | – |

do find that it runs remarkably quickly (6 times faster or more than XGBoost and RandomForest), with a very low-memory footprint considering that it exhaustively captures a full space of alternatives. Thus, we believe that STAND offers a potentially valuable starting point for future big-data-oriented model development, even though that was not our main objective with this work.

*Table 4.* Test Accuracy (%) by Dataset and Model, and average accuracy across models.

| Dataset / Model | DecisionTree | RandomForest | XGBoost | STAND | STAND (heir) |
|---|---|---|---|---|---|
| breast-cancer | $76.66 \pm 0.70$ | $72.24 \pm 0.86$ | $76.66 \pm 0.70$ | $\mathbf{77.14} \pm 0.70$ | $71.97 \pm 0.75$ |
| hepatitis | $80.25 \pm 1.06$ | $\mathbf{84.50} \pm 0.85$ | $78.06 \pm 0.96$ | $80.19 \pm 0.95$ | $81.88 \pm 0.85$ |
| soybean | $90.88 \pm 0.33$ | $\mathbf{93.55} \pm 0.30$ | $90.13 \pm 0.33$ | $91.12 \pm 0.35$ | $93.20 \pm 0.26$ |
| tic-tac-toe | $95.49 \pm 0.24$ | $\mathbf{99.19} \pm 0.09$ | $95.45 \pm 0.25$ | $94.43 \pm 0.25$ | $99.01 \pm 0.10$ |
| vote | $95.82 \pm 0.28$ | $92.91 \pm 0.39$ | $95.73 \pm 0.29$ | $95.82 \pm 0.28$ | $\mathbf{96.36} \pm 0.28$ |
| zoo | $95.64 \pm 0.61$ | $95.64 \pm 0.57$ | $95.45 \pm 0.57$ | $\mathbf{96.36} \pm 0.55$ | $94.64 \pm 0.75$ |
| Average | $89.12 \pm 0.58$ | $\mathbf{89.67} \pm 0.67$ | $89.51 \pm 0.71$ | $88.58 \pm 0.61$ | $89.18 \pm 0.57$ |

*Table 5.* Number of tree nodes for each model by number of samples trained on for very large noisy datasets.

| Dataset / Model | # Samples | OptionTree | DecisionTree | STAND ($\alpha = 0.0$) | STAND ($\alpha = 0.1$) | STAND ($\alpha = 0.2$) |
|---|---|---|---|---|---|---|
| satimage | 200 | 1951 | 47 | 63 | 79 | 108 |
| | 1600 | 106795 | 333 | 431 | 481 | 515 |
| | 5787 | 119195 | 947 | 1184 | 1369 | 1707 |
| letter | 200 | 263547 | 185 | 231 | 392 | 566 |
| | 1600 | 1128373 | 881 | 1366 | 1974 | 3159 |
| | 18000 | 209935 | 4153 | 4778 | 5810 | 7733 |
| isolet | 200 | (too large) | 89 | 98 | 98 | 150 |
| | 1600 | (too large) | 383 | 521 | 552 | 577 |
| | 7018 | (too large) | 1179 | 1510 | 1720 | 2209 |

