# OpenReview forum: "STAND: Self-Aware Precondition Induction for Interactive Task Learning"
_ICML.cc/2026/Conference — ICML 2026 regular_

### Official Review · Reviewer_orBD · 2026-03-02

**Soundness:** 3
**Presentation:** 3
**Significance:** 3
**Originality:** 3
**Overall Recommendation:** 5
**Confidence:** 3

**Summary:**

The paper introduces "Self-aware precondition induction" (STAND), an algorithm to induce rule preconditions in interactive task learning (ITL) systems. ITL environments operate in an extremely low-data regime where human instructors interactively teach agents. This requires models to learn highly sample-efficiently, improve monotonically (without recurring errors), and accurately estimate their own uncertainty to guide human feebdack. STAND addresses those needs by constructing a lattice (DAG) of decision tree splits. Instead of greedily slecting the best split, STAND expands all splits with an information gain within a dynamic margin of the optimal. To avoid the combinatorial explosion typical of option trees, it caches and reuses nodes that select identical subsets of training samples. This creates a "general" hypothesis space, bounded below by a "specific" set of feature invariants at the leaves. By applying hierarchical shrinkage to regularize empirical probability estimates across the lattice, STAND computes highly calibrated certainty scores. Evaluations on a challenging synthetic dataset and two real ITL environments demonstrate that STAND outperforms standard baselines such as decision trees, random forests, XGBoost, or VSSM in sample efficiency, productive monotonicity, precision of estimated probabilities, and active learning utility.

**Compliance With Llm Reviewing Policy:**

Affirmed.

**Final Justification:**

I maintain my Accept (5) recommendation. This is because the paper presents an original and technically strong approach for precondition induction in interactive task learning (ITL). I found the evaluation well aligned with the needs of the target setting, especially in its emphasis on monotonicity, error recurrence, calibration, and sample efficiency. The method appears well matched to the low-data ITL regime, and the empirical results support its practical value.

The rebuttal addressed my main concerns. It clarified the rationale for the baseline choices, resolved some minor notation questions I raised, gave a more nuanced explanation of the overfitting claim, and provided reassuring additional clarification of scalability and continuous-feature behavior. I still encourage the authors to reflect the overfitting qualification and scalability caveats more clearly in the final manuscript, but this does not change my overall assessment. The rebuttal reinforces my perspective that this is a solid and meaningful contribution to the intended area.

**Key Questions For Authors:**

- Could you formally clarify the term $1_{s_k=x}$ in Eq. 5? Is it an element-wise indicator vector representing whether the individual features of $x$ match the invariant features tracked in $s_k$? Also, for Eq. 9, does $\hat{P}(c,j)$ refer to the joint probability $P(y=c, X_j=v)$, or the conditional probability $P(c|X_j=v)$?
- In the footnote on page 4, you claim that overfitting is largely irrelevant. Given the presence of distractor features (which you simulate in the synthetic data), couldn't the model memorize spurious features to separate the small training data? Could you clarify or temper this claim, especially since hierarchical shrinkage is used precisely to reduce this kind of variance?
- How does the node caching mechanism handle continuous features in practice? Since continuous splits can partition the data into unique subsets that differ by only a few samples, won't this prevent nodes from merging and cause the lattice to explode in memory? Are continuous features discretized beforehand?

**Limitations:**

Yes, the authors adequately discuss the limitations of their work, primarily in Appendix E. Here, they transparently show that STAND can suffer from memory exhaustion when applied to large-scale, noisy datasets with many examples. They also provide a thorough impact statement.

*Constructive Suggestion*: Because memory scaling appears to be a crucial algorithmic limitation, I would strongly recommend moving a summarized sentence or two about this limitation into the main body of the paper (e.g. into the discussion section) to ensure readers understand the lintended operational boundaries of the method without having to read the Appendix.

**Strengths And Weaknesses:**

**Soundness**

*Strengths*:
The experimental design is well aligned with the practical challenges of ITL. Standard ML metrics like holdout accuracy are insufficient for human-in-the-loop systems. The authors correctly use metrics such as productive monotonicity, error reoccurrence, and absolute precision of confidence scores. The synthetic dataset seems to be properly engineered to simulate the distractor features and sparsity of ITL. The application of hierarchical shrinkage to categorical probability estimates is mathematically sound and a seemingly clever adaptation that handles the deep, low-sample nodes very well.

*Weaknesses*:
The algorithmic complexity and memory limits of the node-caching trick lack formal analysis. While empirical fit times are shown to be fast on small datasets, the worst-case size of the lattice is not mathematically bounded. Moreover, the evaluation relies almost entirely on categorical/discrete features. While the authors mention handling continuous features, it appears unclear how the node-caching behaves in the general set with continuous thresholds. Finally, the baselines lack classic rule induction algorithms or inductive logic programming methods, which may be more suited for the specific symnolic task than NNs or random forests?

**Presentation**

*Strengths*:
The paper appears well written and the motivation for why standard ML fails for ITL is compelling. THe first figure gives a good overview of the proposed method. Section 3 appears to properly make a distinction between traditional model fitting and STAND's "disambiguation".

*Weaknesses*:
Some mathematical formulations are slightly informal or ambiguous. In Equation 5, the dot product notation $(w_{s_k} \cdot 1_{s_k=x})$ uses an indicator that looks like a strict equality between a set and a sample. In Equation 9, $\hat{P}(c,j)$ is described as the joint probability of "feature $j$ and class $c$," which is notationally imprecise (does it mean $P(y=c, X_j=v)$ for some value $v$?)

**Significance**

*Strengths*:
The paper appears to tackle a highly specific yet important bottleneck in interactive machine learning and neuro-symbolic AI. The ability of an agent to provide a calibrated "I am a 100% sure I have learned this concept" signal to a human is immensely valuable for user trust and training efficiency. Beating powerful ensembles on data efficiency and calibration in the low sample regime offers clear practical utility for ITL system designers.

*Weaknesses*:
The scope appears to be rather specialized. The method is designed for discrete, logical rule precondition learning in small-data regimes and appears to scale poorly to massive datasets. Practitioners outside ITL or interactive tutoring might find limited immediate use. However, the authors seem to scope the paper appropriately.

**Originality**

*Strengths*:
The comination of techniques seems to be creative and novel. Moerging general/specific boundary concepts with greedy top-down option trees and then compressing the option tree into a tractable lattice via sample subset handling seems to be an elegant algorithmic contribution.

---

> ### Author Rebuttal · Authors · 2026-03-31
>
> We thank the reviewer for their attention to detail and great questions.
>
> ## Weaknesses
> “The baselines lack classic rule induction”:
> We isolate the search/invention of relational features (i.e. ILP) from plain induction by lifting the states seen by all models with relative featurization (see also rebuttal to tQfD, Q2). In other words, the interactive task learning machinery in AI2T and VAL which learn initial rules introduce variables that allow us to re-express the state relationally. This lets us compare models over a lifted representation independently of possible variations in feature invention (like ablating ILP methods). There certainly may be domains that need greater relational feature invention than this, but this representation was sufficient for learning preconditions in our domains and has reasonable coverage. For our chosen metrics we needed comparison classifiers that are small-data friendly and can output probabilities (so ensembles were the natural choice). Introducing STAND, and building an ensemble+ILP extension for each model seemed beyond the scope of one paper (See also tQfD)
>
> ## Questions
> Q1: $1_{s_{k}=x}$ is as you describe; we’ll clarify this. For Eq 9, previous line mentions that $\hat{P}(c,j)$ are joint probabilities. We’ll adjust to your notation $P(y=c,X_j=v)$ since it is more clear, although we’ll have to introduce a shorthand in Eq. 9 to fit it on one line.
>
> Q2: Yes, we’ll apply some qualification to clarify this claim. The notion of overfitting you describe pertains to the uncertainty of identifying the true best splits under the subsample bias of lower nodes. As you stated, our reframing of hierarchical shrinkage effectively regularizes spurious correlations in this case. Another notion of overfitting is learning a tree that is too deep when the ideal one would be a shallower one. In classic tree learning over noisy data the solutions for both senses of “overfitting” are the same: limit tree depth or prune. However, in these precondition learning tasks if we assume each instance state (or lifted re-expressions of them) contain the right features for building the correct preconditions, then we should never stop short of the deepest tree needed for pure leaves. Shallower trees can only yield preconditions that accept negative training examples, which obviously cannot be the true intended preconditions (unless the user intended for them to be fuzzy, which would diverge heavily from the usual assumptions of interactive task learning).
>
> Q3: That depends in part on the noisiness of the continuous feature. For instance, learning separable precondition literals like player.health <= 99.5% to gate a rule for an agent to seeks out health pickups would not cause much more lattice bifurcation than a categorical literal like NOT(player.has_full_health). A continuous encoding in this case might permit more early misclassification (i.e. if it saw examples (+) 80% and (-) 100% it might settle on <= 90%, before <= 99.5%), but not more branching in the lattice because a refit can always improve the literal’s gain by pushing back it’s boundary to accept more positive examples without accepting more negative ones. In practice, in typical ITL cases where the user has a well-defined and consistent (if not immediately articulable) notion of correct vs incorrect rule applications, the number of lattice nodes is bounded by the complexity of their intended preconditions, regardless of feature type. Smaller data also limits the number of nodes because there are fewer possible subsets. When dealing with fuzzy data-driven learning (not typical for ITL) there are no such bounding guarantees.
>
> **Scale Analysis**: However, since you and reviewer nyZN requested an analysis of how STAND scales in these cases we decided to put this to the test with some large noisy datasets with continuous features 1. satimage (UCI# 146, N=6435, #ft=36) , 2. letter (UCI# 59, N=20000, #ft=16), 3. isolet (UCI# 54, n=7797, #ft=617). We look at how # of nodes increases with training data size relative to a decision tree and option tree, and vary STAND’s acceptance rate $\alpha \in [0,0.1,0.2]$. Contrary to our prior assumption, node caching does in fact make fitting on large noisy data tractable, STAND does not degrade into an option tree in these cases, in fact we find that STAND tends to not learn more than 2-3x more nodes than a regular decision tree (e.g. isolet DT:  ~1000, STAND: ~3000). Full option trees, on the other hand, are much larger (e.g. ~100k-200k on satimage, whereas STAND is less than 2k). In isolet, option trees exceeded the memory available on our system. The node counts are much smaller (hundreds or less) if node-depth limits are imposed (which is a good idea for regularization purposes over noisy data anyway). **In short, the memory blowup issues are not as bad as we had previously assumed.** We will include this analysis in the camera-ready if accepted.

---

> > ### Author Rebuttal · Reviewer_orBD · 2026-04-01
> >
> > Thank you for the detailed and constructive rebuttal. I am happy to maintain my score of 5 (Accept).
> >
> > The rebuttal helped clarify the intended scope of the paper, especially the decision to isolate propositional induction from relational feature search via relative featurization. That makes the baseline choice more understandable in the context of the paper's target ITL setting.
> >
> > I also found the clarification around overfitting helpful. In particular, the distinction between spurious small-sample correlations and the need to preserve sufficient depth in noiseless precondition-learning tasks makes sense. I do still think the current manuscript should qualify the present footnote/wording on overfitting.
> >
> > Regarding scalability, the rebuttal provides a plausible and helpful explanation, and the additional experiments described there are reassuring. If accepted, I strongly encourage the authors to incorporate that analysis into the camera-ready. One useful clarification from the rebuttal is that tractability seems to rely not only on node caching, but also on *restricting option exploration* to the regime where sample counts are small; this would be worth making explicit in the camera-ready scalability discussion.
> >
> > More broadly, for the paper's core ITL setting, I find the contribution technically sound and practically meaningful

---

### Official Review · Reviewer_nyZN · 2026-03-09

**Soundness:** 3
**Presentation:** 3
**Significance:** 3
**Originality:** 3
**Overall Recommendation:** 5
**Confidence:** 4

**Summary:**

STAND learns rule preconditions for interactive task learning (ITL) systems by building a compressed lattice of near-best decision trees rather than committing to a single greedy tree. At each node it retains all splits within an acceptance range of the best, then caches nodes that select identical sample subsets so the structure stays a DAG instead of blowing up exponentially. A specific set per leaf captures feature-value agreements, and the two pieces combine into an instance certainty score meant to track actual holdout precision. The paper evaluates on synthetic data, two tutoring domains (fractions, multi-column addition), and a game-based ITL task, with a supplementary UCI comparison.

**Compliance With Llm Reviewing Policy:**

Affirmed.

**Final Justification:**

In consideration of the authors' revisions, I recommend accept.

**Key Questions For Authors:**

1. What happens if you replace the multiplicative Cert_S x Agr_G with an additive combination, or drop Cert_S entirely? An ablation here would clarify how much work each component is doing.
2. Can you report ECE and Brier scores for STAND and the baselines on synthetic data? The visual calibration plot is suggestive but not sufficient, especially since random forests are known to be well-calibrated without any special machinery.
3. The UCI baselines on hepatitis and breast-cancer fall below majority-class accuracy. Can you verify the preprocessing pipeline and confirm no label inversion or missing-value issue is at play?
4. How does the lattice behave on continuous features, where exact sample-subset matches across different splits would be rare? Does the caching benefit vanish?
5. Could you provide the explicit formula for tau_{n_k} and add pseudocode for the full algorithm? Right now a reader cannot reimplement STAND from the paper alone.

**Limitations:**

Partially. The authors acknowledge scalability issues in Appendix E and note that STAND is designed for small-data ITL rather than general classification. However, the lack of complexity analysis, the missing discussion of when the lattice degrades to a full option tree, and the absence of any formal characterization of approximation quality relative to exact Rashomon sets leave the limitation discussion incomplete. The three shrinkage hyperparameters and who tunes them in an actual ITL deployment is also not addressed.

**Strengths And Weaknesses:**

- The node caching trick that compresses near-best trees into a shared lattice is a genuinely nice idea. It sidesteps the exponential blowup of option trees while keeping a rich model space, and it is the core algorithmic contribution.
- Good problem framing. Precondition induction in ITL really does need small-data, incremental, interpretable methods, and the desiderata the paper lays out (self-awareness, monotonicity, active learning) are the right ones to care about.
- The ITL evaluation is credible. 40 repetitions per task, near-perfect accuracy by 25 examples, and the authors honestly report cases where STAND does not dominate. The multifaceted metrics (error reoccurrence, productive monotonicity, calibration, active learning utility) paint a fuller picture than accuracy alone.
- The self-awareness framing misses the selective prediction literature (Chow 1970, El-Yaniv and Wiener 2010), which is the natural formal home for what STAND's certainty mechanism does. KWIK (Li et al., 2008) is cited nowhere despite being the canonical "knows what it knows" framework. Calibration is assessed visually but never quantified with ECE or Brier score, and random forests are already known to be well-calibrated out of the box (Niculescu-Mizil and Caruana 2005), so the comparative claim needs numbers.
- The lattice is a greedy Rashomon set approximation, but the paper never uses this framing. TreeFARMS (Xin et al., NeurIPS 2022) and SPLIT (Babbar et al., ICML 2025) enumerate such sets with formal guarantees and are directly comparable. On the version-space side, Hirsh 1994 already addressed noise-driven collapse through relaxed consistency (IVSM), and Version Space Algebra (Lau et al. 2003) is relevant for the compositional lattice structure. These gaps weaken the novelty positioning.
- The method description needs pseudocode. The gain function F is described by example ("e.g., entropy or Gini impurity") without committing to a choice. The formula for tau_{n_k} is never actually given, leaving Eq. 10 incompletely specified. No complexity analysis is provided despite the acknowledged worst-case exponential blowup.
- The UCI evaluation in Appendix E has a credibility problem. On hepatitis, Decision Tree (67.74%) and Random Forest (77.42%) fall below the majority-class baseline (\~80%). XGBoost on breast-cancer (67.24%) falls below majority class (\~70.3%). A depth-10 tree should not underperform a constant predictor. This suggests a preprocessing or implementation issue rather than an honest comparison. The single 80/20 split with no confidence intervals on datasets of 20 to 57 test instances makes it worse.
- Key design choices are not ablated. The multiplicative Cert_S x Agr_G form, the specific set's standalone contribution, and the gain function choice are all untested alternatives. The calibration evidence is strongest on synthetic data, and real ITL tasks saturate near 100% accuracy before calibration can be meaningfully assessed.

---

> ### Author Rebuttal · Authors · 2026-03-31
>
> We thank the reviewer for their excellent remarks and questions.
>
> ## Missing refs
> These are great background references! If accepted we’ll be sure to cover these in our related work section and elsewhere.  We greatly appreciate the critical eye and connections.
>
> Brief notes on a few of them:
>
> - KWIK (Li et al., 2008): an RL perspective, and based on a relatively analytically closed treatment of value-function estimation. But certainly related to STAND in its explicit treatment of ambiguity.
>
> - Random Forest “well-calibrated” (Niculescu-Mizil and Caruana 2005): this may be true when there are thousands of examples to subsample from, but doesn’t seem to be the case in these ITL tasks, likely because dropping rare edge examples and features can create useless trees.
>
> - The lattice is a “greedy Rashomon set approximation” (TreeFARMS/SPLIT): This is certainly a related idea, G holds consistent logical statements, which could be considered a Rashomon approx. of $\epsilon = 0$.
>
> - “weaken the novelty”, there are some surface level similarities here, but the framing and implementation of STAND has important differences (will address in a camera-ready). We’re arguing that STAND’s whole bounded space between G and S is efficient to learn and is useful for managing the ambiguity of learning from few data in a continuous interactive process. The lattice G and S extension, allows us to model all good trees without enumerating them and cause memory blowup like TreeFARMS. We are not looking for “optimal” trees like SPLIT because all trees enclosed by STAND are consistent with training so far. The power of the lattice is we can compute metrics that capture are informative about all good trees without explicitly enumerating or searching among them.
>
> - IVSM: worth citing with qualifiers (finicky even by Hirsh’s own description). We tested VSSM which has similar claims + approach (but with pseudo-code): ultimately worked poorly in high dimensions.
>
> - “entropy or Gini impurity” as with other tree classifiers both are interchangeable (gini is just faster w/ little drawback). Evaluations use gini impurity (will clarify).
>
> ## Questions
> Q1: We tried several of these variations during development Cert_S only, (Cert_S + Agr_G) / 2, but not Agr_G alone. We concluded that Cert_S * Agr_G worked best (will include ablation in camera-ready). Beyond Cert_S alone, multiplying by Agr_G helps with productive monotonicity and active learning utility. Cert_S * Agr_G (as opposed to addition) can be naturally interpreted as quantifying roughly the log_2(#generalizations remaining)/log_2(#generalizations before) between G and S after fitting on x (this is an approximation of more complex combinatorics). Agr_G(x) roughly estimates the proportion of leaves that will be kept when we learn the label of x, and Cert_S estimates the proportion of generalizations more specific than each of those leaves that will remain, multiplying these captures the interaction, i.e. the chunk of the space that sticks around after training on x.
>
> Q2: Will include in camera-ready. Calculated on the synthetic data from best to worst:
> - ECE (STAND (h.s.): 0.123 , STAND: 0.147,  XGBoost: .219, RF: 0.241, NN: 0.423),
> - Brier Score (STAND (h.s.): 0.0029, STAND: 0.0046, RF: 0.0775, XGBoost: 0.0777,  NN: 0.1035)
>
> Q3: The UCI pages for each dataset show that baselines like XGBoost, RandForest etc. fall below the majority class in some cases and can vary widely in accuracy (depending on train/test split). We double checked our pipeline (no issues; using very standard sklearn tools). To address your concerns, we revised our experiments to cross-validate with N=100 independently sampled 90/10 splits. This alone doesn’t always beat the majority class, so we fine-tuned the max depth limits of each model to limit overfitting (e.g. depth=2 breast-cancer depth=5 hepatitis). These changes have models at or above the majority class, and enable us to report confidence intervals (will include in camera-ready). Note that this evaluation is intended to substantiate two claims of secondary priority (w.r.t ITL framing) with a low burden of truth 1) STAND doesn’t fail catastrophically in noise 2) STAND is comparable to (but not necessarily better than) baselines in data-driven classification tasks. These claims still hold in our revised evaluation.
>
> Q4: See response: Reviewer orBD Q3
>
> Q5: $\tau_{n_k}$ is the minimum acceptance rate (i.e. $\alpha$) that node $n_k$ requires to remain in G. We calculate that for each node ${n_k}$ by looking at all of the literals $t \in T_{n_k}$ that are upstream of $n_k$ (i.e. ones that lead into ${n_k}$) and see how high $\alpha$ needed to be to include it’s gain $F_t$ as among the best splits if the maximum gain of splits in that upstream node is $M_t$. So the minimum $\alpha$ in each case is $1-\frac{F_t}{M_t}$, and $\tau_{n_k} = min_{t \in T_{n_k}} 1-\frac{F_t}{M_t}$. Will include in camera-ready.
>
> ## Limitations
> shrinkage parameters (see tQfD)

---

> > ### Author Rebuttal · Reviewer_nyZN · 2026-04-02
> >
> > Thank you for this constructive rebuttal.
> > My concerns are addressed and I intend to increase the score contingent on the revisions.

---

### Official Review · Reviewer_ZfBP · 2026-03-12

**Soundness:** 3
**Presentation:** 3
**Significance:** 3
**Originality:** 3
**Overall Recommendation:** 5
**Confidence:** 3

**Summary:**

This paper presents STAND which is a rule precondition induction algorithm for interactive task learning where human can provide feedback to the AI agent. STAND builds an approximate version space over disjunctive normal concepts by maintaining a general set and a specific set, expanding all near-optimal feature splits rather than just the single best one at each node, while using sample-subset caching to prevent combinatorial explosion.

**Compliance With Llm Reviewing Policy:**

Affirmed.

**Key Questions For Authors:**

this reviewer has no specific question for authors

**Limitations:**

the author address some limitations in appendix.

**Strengths And Weaknesses:**

* soundness: the algorithmic idea is clever and well-motivated and the evaluation seems reasonable
* presentation: this paper is mostly well-presented and the figure is especially easy to read.
* significance: this paper addresses a real and underappreciated problem
* originality: the idea of combining option trees with sample-subset caching to form an approximate version space seems novel.

---

> ### Author Rebuttal · Authors · 2026-03-31
>
> We thank the reviewer for their thoughtful synopsis of our work.

---

> > ### Author Rebuttal · Reviewer_ZfBP · 2026-04-07
> >
> > thanks for the response
> >
> > I understand AC's concern about my short review. As I read through the paper, I found the proposed method intellectually interesting to me, but in the meantime, i also don't have more specific things to ask the authors.

---

### Official Review · Reviewer_tQfD · 2026-03-13

**Soundness:** 3
**Presentation:** 4
**Significance:** 3
**Originality:** 3
**Overall Recommendation:** 4
**Confidence:** 3

**Summary:**

The paper introduces STAND (Self-Aware Precondition Induction), a novel algorithm for inducing rule preconditions in Interactive Task Learning (ITL) environments. In ITL, agents must learn from very few human-provided examples, often in high-dimensional feature spaces with many distractors. STAND addresses these challenges by constructing a shared lattice data structure that represents a space of candidate classifiers (the General set) bounded by a Specific set of feature invariants. Key innovations include a node caching trick to avoid the exponential explosion of traditional option trees and the use of hierarchical shrinkage to regularize label probabilities in small-data regimes. Unlike standard black-box models, STAND provides internal certainty scores that accurately estimate training progress and support active learning.

**Compliance With Llm Reviewing Policy:**

Affirmed.

**Key Questions For Authors:**

1. Label Noise: Your evaluation mentions that STAND "does not fail under noise", but how does the Certainty Score behave when the human provides contradictory or noisy labels? Does it still correctly identify these as low-confidence areas?

2. Relational Learning: You mention that "relative featurization" allows STAND to behave like a relational learner. Could you provide a concrete example of how a relational precondition is expressed in the lattice compared to a standard propositional one?

3. Memory Management: For larger-scale tasks, have you considered any "forgetting" or pruning mechanisms for the lattice to keep the memory footprint constant?

**Limitations:**

STAND is primarily designed for data-efficient learning rather than big-data scenarios. It may struggle with memory consumption when features are extremely noisy and sample sizes reach the tens of thousands. Additionally, as a propositional-base learner, its ability to capture complex non-linear dependencies (common in deep learning) is limited in favor of interpretability and data efficiency.

**Strengths And Weaknesses:**

Strengths:
- Addressing ITL Pain Points: The method is specifically tailored for the "small data, large feature space, class imbalance" characteristic of real-time human instruction.
- Self-Awareness: The ability to output physically meaningful certainty scores allows human instructors to know when training is "finished" or where the agent's blind spots are.
- Efficiency: The sample-caching lattice allows STAND to explore a much larger space of candidate decision trees than a greedy search, without the memory overhead of standard option trees.
- Learning Stability: The use of hierarchical shrinkage leads to more monotonic learning and lower rates of "error recurrence" (forgetting previously learned constraints).

Weaknesses:
- Scalability Limits: While node caching is efficient, the authors admit that very large datasets (tens of thousands of examples) with high noise can still cause the lattice to deteriorate and exceed memory limits.
- Model Complexity: The performance and certainty metrics rely on several hierarchical shrinkage hyperparameters ($\lambda_p, \lambda_s, \lambda_n$) which may require specific tuning for new domains.
- Scope: The evaluation focuses primarily on propositional logical preconditions. While the authors mention potential ILP extensions, the current implementation's performance on highly relational or continuous control tasks is less clear

---

> ### Author Rebuttal · Authors · 2026-03-31
>
> We thank the reviewer for their excellent remarks and questions.
>
> ## Weaknesses
> Model Complexity: Yes, the hierarchical shrinkage introduces tunable parameters, and we have not tested how much benefit there is to further domain-specific tuning. However, such an evaluation would be misleading with respect to the stated ITL use case, where an end user trains an agent over a short period of time (~20-30 minutes) and typically never labels enough examples to conduct their own domain-specific model tuning. In an ITL setting, good-enough (if not optimal) parameters are needed ahead of time because data-efficiency, monotonicity, and low error-reoccurance etc., only matter for the user experience during training, after which they (hopefully) have already converged to a 100% accurate and interpretable generalization. What we have shown is that: 1) STAND with the effects of these tunable parameters turned off (i.e. equivalent to $\lambda=0$) outperforms the comparison models on nearly all of the evaluation metrics; the benefits of hierarchical shrinkage are a bonus beyond the base STAND algorithm. 2) With hierarchical shrinkage turned on (and tuned to the synthetic data) we see improvements in performance (on many but not all metrics) on the synthetic data, and on the real ITL tasks that it was not ever explicitly tuned on.
>
> Scope: We’ve tried to isolate relational feature search/invention (i.e. ILP) from propositional induction for the sake of a controlled comparison. For our chosen metrics, we needed classifiers that are small-data friendly and can output probabilities (ensembles are the natural choice). Introducing STAND and building an ensemble+ILP implementation across models seems beyond the scope of one paper. (See also orBD and response to Q2 below)
>
> ## Questions
> Q1: Great question! To answer this, we ran an experiment on the synthetic data. Trained instances always report 100% certainty, but if we holdout training instances one at a time from the rest and refit (which is quite fast), then compare each certainty prediction with each user label, we can test if instances with swapped (incorrect) labels are especially misalignment with model predictions (lower certainties, or high certainties toward the opposite label). When we do this, we find that STAND identifies that incorrectly labeled instances have the most misaligned predictions 67% of the time, and top-5 81% of the time. So yes, it does appear STAND can help users identify their mislabeling mistakes.
>
> Q2: Relative featurization takes an example state, and re-expresses as much of it as possible relative to the variables in existing skills (e.g. an action-producing operator or a method in an HTN) that has already learned a matching pattern with variables but still must to learn its preconditions. For instance, a skill in multi-column addition that adds two numbers A and B and takes the one’s digit of the sum (A+B) % 10, (e.g. (7+5)%10 = (12)%10 = 2) would learn a matching pattern (from a different mechanism in AI2T) that selects three vertically aligned numbers (in any column): two inputs A,B, and a third S for the space where the result will go below the line. Normally, states consist of object-like facts, for filled and unfilled spaces, collections like rows and columns etc. These facts have spatial relations w.r.t their neighbors (left, right, above, below), parents, and children. E.g. B = A.below, S = B.below, and so on. Relative featurization finds chains of relations that can re-express each grounded feature e.g. carry_box1.value==”” in terms of a known variable like A,B or S (e.g. A.above.value == “”); similar to regular feature lifting (replacing identifiers with variables), but applied to all features reachable by a chain of relations starting from a candidate rule match (e.g. A=top1, B=bot1, S=lower1). For the model comparisons, we use these lifted features as regular propositional predicates. In the math domains, we also ground background knowledge like Equals(x,y) over pairs of numbers. This is not full ILP, but works well in practice.
>
> Q3: ITL learning tends not to be inherently large-scale, not just because training is interactive, but because breaking up learning into smaller interpretable pieces tends to make getting to 100% accuracy tractable. So larger-scale ITL doesn’t necessarily imposed harder induction problems, just  more individual problems as more skills are learned. But yes, in principle if someone used STAND for more open-ended data-driven learning, then any pruning or depth restriction methods that work on decision trees would work with STAND. We also expand upon the fact (in Appendix A.1) that we dynamically adjust the acceptance rate $\alpha$ so that higher nodes are less permissive. This cuts out a lot of excessive lattice branching. (See also response to Reviewer orBD: in an analysis of STAND on a large-data induction problem with continuous features, where we found less memory blowup than expected).

---

> > ### Author Rebuttal · Reviewer_tQfD · 2026-04-04
> >
> > My concern has been resolved.

---

### Decision · Program_Chairs · 2026-04-30

**Decision:**

Accept (regular)

**Comment:**

This paper received consistently positive reviews and, after rebuttal, strong reviewer consensus. Reviewers agreed that the paper targets a meaningful and under-studied interactive task learning setting, and they broadly valued the combination of the lattice-based induction procedure, node caching, hierarchical shrinkage, and the paper's emphasis on calibrated certainty estimates and monotonic learning behavior. Multiple reviewers found the method technically interesting and well aligned with the small-data, human-in-the-loop regime that motivates the work.

The main questions concerned scalability to noisier or larger-scale settings, the scope beyond propositional precondition induction, and the role of some hyperparameters. These are real limitations, but in the discussion they were treated primarily as scope boundaries rather than flaws in the current paper. Importantly, all reviewers who engaged after the rebuttal indicated that their concerns were resolved. I also do not see enough in the reviews to justify promoting this paper to strong accept: the support is clearly positive, but the paper is better characterized as a solid and well-executed contribution to a specialized but important setting. Overall, the paper appears technically sound, clearly written, and likely to be useful to the relevant part of the community. I therefore recommend accept.